# Receptor-mediated clustering of FIP200 bypasses the role of LC3 lipidation in autophagy

Amelia E Ohnstad[1], Jose M Delgado[1], Brian J North[1], Isha Nasa[1,2] (iD), Arminja N Kettenbach[1,2] (iD), Sebastian W Schultz[3,4] & Christopher J Shoemaker[1,*] (iD)

## Abstract

Autophagosome formation requires multiple autophagy-related (ATG) factors. However, we find that a subset of autophagy substrates remains robustly targeted to the lysosome in the absence of several core ATGs, including the LC3 lipidation machinery. To address this unexpected result, we performed genome-wide CRISPR screens identifying genes required for NBR1 flux in ATG7[KO] cells. We find that ATG7-independent autophagy still requires canonical ATG factors including FIP200. However, in the absence of LC3 lipidation, additional factors are required including TAX1BP1 and TBK1. TAX1BP1's ability to cluster FIP200 around NBR1 cargo and induce local autophagosome formation enforces cargo specificity and replaces the requirement for lipidated LC3. In support of this model, we define a ubiquitin-independent mode of TAX1BP1 recruitment to NBR1 puncta, highlighting that TAX1BP1 recruitment and clustering, rather than ubiquitin binding per se, is critical for function. Collectively, our data provide a mechanistic basis for reports of selective autophagy in cells lacking the lipidation machinery, wherein receptor-mediated clustering of upstream autophagy factors drives continued autophagosome formation.

**Keywords** ATG7; autophagosome; NBR1; selective autophagy; TAX1BP1
**Subject Categories** Autophagy & Cell Death; Membranes & Trafficking
**The EMBO Journal (2020) 39: e104948**

See also: **TN Nguyen & M Lazarou** (December 2020)

## Introduction

Macroautophagy (hereafter autophagy) is a cellular trafficking pathway that delivers cytoplasmic components—such as damaged organelles, protein aggregates, and bulk cytoplasm—to the lysosome for degradation. During autophagy, a specialized double-membrane vesicle, the autophagosome, is generated *de novo* around cytoplasmic cargo (Søreng *et al*, 2018; Kirkin, 2020). Completed autophagosomes are subsequently trafficked to the lysosome where their cargoes are degraded by lysosomal hydrolases. Consequently, defects in autophagy result in the accumulation of toxic species and are implicated in the etiology of many diseases including neurodegeneration, cancer, and aging (Choi *et al*, 2013; Levine & Kroemer, 2019).

Autophagosome biogenesis requires the coordinated activity of multiple autophagy-related (ATG) proteins. Early steps in autophagy are marked by recruitment of the FIP200 scaffold protein and ATG9-containing vesicles to sites of autophagosome formation (Itakura & Mizushima, 2010; Koyama-Honda *et al*, 2013; Kishi-Itakura *et al*, 2014). Atg8-family proteins, including both the LC3 and GABARAP families in mammals, are subsequently conjugated onto the expanding autophagosomal membrane via a ubiquitin-like conjugation cascade requiring ATG7 (E1-like), ATG3 and ATG10 (E2-like) and ATG5/ATG12/ATG16L1 (E3-like) (Ichimura *et al*, 2000; Geng & Klionsky, 2008). Mammalian Atg8 homologs (hereafter LC3) are implicated in many steps in autophagosome formation including cargo selection (Birgisdottir *et al*, 2013; Sawa-Makarska *et al*, 2014; Stolz *et al*, 2014), membrane expansion and closure (Nakatogawa *et al*, 2007; Fujita *et al*, 2008; Sou *et al*, 2008; Weidberg *et al*, 2010; Kishi-Itakura *et al*, 2014; Tsuboyama *et al*, 2016), vesicle trafficking (Manil-Ségalen *et al*, 2014; Nguyen *et al*, 2016; Vaites *et al*, 2017; Gao *et al*, 2018), and degradation (Tsuboyama *et al*, 2016).

Despite the many functions ascribed to LC3, accumulating evidence indicates that selective autophagy can operate in its absence. Pioneering studies found that mitochondrial clearance during erythroid maturation is mediated by double-membrane autophagosomes in an ATG5/7-independent manner (Nishida *et al*, 2009; Honda *et al*, 2014, 7). Similarly, PINK1/Parkin-dependent mitophagosome formation was found to persist in cells lacking all Atg8-family proteins, although the resulting autophagosomes were smaller, formed less efficiently, and could not fuse with lysosomes (Nguyen *et al*, 2016). Yet another alternative route to the lysosome was observed for NCOA4, which enables the degradation of ferritin (a cytoplasmic, iron-binding protein aggregate) (Goodwin *et al*, 2017). With the exception

1   Department of Biochemistry and Cell Biology, Geisel School of Medicine at Dartmouth, Hanover, NH, USA
2   Norris Cotton Cancer Center, Lebanon, NH, USA
3   Centre for Cancer Cell Reprogramming, Faculty of Medicine, University of Oslo, Oslo, Norway
4   Department of Molecular Cell Biology, Institute for Cancer Research, Oslo University Hospital, Oslo, Norway
    *Corresponding author. Tel: +1 603 650 1725; E-mail: christopher.j.shoemaker@dartmouth.edu

of ATG9A, ferritin turnover persisted in the absence of virtually all other known ATG factors, including the LC3-lipidation machinery (Goodwin *et al*, 2017). In sum, these discoveries are beginning to reshape the monolithic view of how ATG factors function into a more complex web of overlapping mechanisms that ensure robust cytoplasm-to-lysosome delivery.

Here, we further explored routes to the lysosome by focusing on the SQSTM1-like family of autophagy receptors, including NDP52, SQSTM1, TAX1BP1, and NBR1 (Birgisdottir *et al*, 2013). SQSTM1-like receptors (SLRs) are soluble, cytosolic proteins that, individually or in combination, bind autophagic cargoes and mark them for degradation. SLRs share at least three defining features: a ubiquitin-binding domain, an LC3-binding motif, and an oligomerization domain (for reviews, see (Kirkin & Rogov, 2019; Johansen & Lamark, 2020)). Accordingly, SLRs perform related, if not redundant, functions during many forms of selective autophagy, including mitophagy (Lazarou *et al*, 2015), xenophagy (Tumbarello *et al*, 2015), and aggrephagy (Kirkin *et al*, 2009; preprint: Sarraf *et al*, 2019). At the same time, receptor diversity confers individual SLRs with non-overlapping functions, such as targeting unique autophagy substrates or interfacing with additional cellular pathways. For instance, NBR1 functions coordinately with SQSTM1 in aggrephagy (Kirkin *et al*, 2009) and pexophagy (Deosaran *et al*, 2013), yet uniquely targets alternative substrates (e.g., major histocompatibility complex class I [MHC-I]) (Yamamoto *et al*, 2020). Additionally, NBR1 is one of several receptors that, in response to acute starvation, also becomes a target of endosomal microautophagy (Mejlvang *et al*, 2018).

Inspired by the pH-sensitive, tandem-fluorescent (tf) reporter for monitoring LC3 lysosomal delivery (tf-LC3) (Kimura *et al*, 2007; Pankiv *et al*, 2007), we previously developed tf-Receptor fusion proteins as phenotypic reporters for genome-wide CRISPR screening (Shoemaker *et al*, 2019). Unexpectedly, we find that lysosomal targeting of NBR1 (and to a lesser extent TAX1BP1 and SQSTM1, but not NDP52) persists in cells lacking LC3 conjugation factors. Subsequent screens performed using lipidation-deficient (e.g., $ATG7^{KO}$) cells enabled us to compare genetic modifiers of lipidation-dependent and lipidation-independent autophagy across the entire genome. We find that lipidation-independent autophagy absolutely requires canonical ATG factors implicated in phagophore nucleation, including ATG9A and FIP200, as well as components of the late endocytic pathway including RAB7A and the homotypic fusion and vacuole protein sorting (HOPS) complex. However, in the absence of LC3 lipidation, additional factors are required for selective autophagy including TAX1BP1 and TBK1. Under this alternative regime, TAX1BP1's ability to cluster FIP200 around NBR1 promotes continued autophagosome formation and enforces selective cargo incorporation. Both the SKICH domain and a newly identified N domain are required for this LC3-independent, UBZ-independent function of TAX1BP1. These data provide a mechanistic basis for reports of selective autophagy in cells lacking the lipidation machinery (e.g., $ATG7^{KO}$ or $ATG5^{KO}$ cells). Furthermore, our data reinforce the duality of mammalian autophagy receptors in both tethering cargo to autophagic membranes (via LC3) and, independently, recruiting upstream autophagy factors to drive local autophagosome formation.

# Results

## Autophagy receptors are differentially regulated in the absence of LC3 lipidation

We previously developed a family of tandem-fluorescent (tf) autophagy reporters derived from the tf-LC3 reporter system (Shoemaker *et al*, 2019). These reporters consist of red fluorescent protein (RFP) and green fluorescent protein (GFP) in frame with any of several soluble autophagy receptors: NDP52, SQSTM1, TAX1BP1, and NBR1 (hereafter tf-Receptors) (Fig 1A). Upon delivery of tf-Receptors to the acidic lysosomal compartment, GFP fluorescence is quenched while RFP fluorescence persists. Thus, red:green fluorescence ratio serves as a quantitative measure of lysosomal delivery with single-cell resolution. When combined with genome-wide CRISPR screening, our reporters enable us to identify genetic modifiers of mammalian autophagy. Using this approach, we previously noted that tf-Receptors are differentially influenced by LC3 lipidation (Shoemaker *et al*, 2019). To extend these initial observations, we transduced our tf-Receptor cell lines with single-guide RNA (sgRNA) constructs targeting ATG7, ATG9A, or a non-targeting control. sgATG7 and sgATG9A equally inhibited tf-NDP52 or tf-LC3. In contrast, tf-NBR1 was minimally affected by sgATG7, retaining > 65% of flux relative to sgATG9A (Fig 1B). tf-SQSTM1 and tf-TAX1BP1 showed intermediate effects.

To ensure that residual NBR1 flux was not due to variable knockout efficiency between cell lines, we generated clonal knockout cell lines using CRISPR/Cas9. These stable cell lines express tf-NBR1 but lack individual autophagy genes (Fig EV1A). Cells lacking ATG9A or FIP200 were fully defective for autophagy as chemical inhibition of autophagy by Bafilomycin A1 (BafA1) resulted in no further decrease of red:green ratio (Fig 1C). In contrast, tf-NBR1 flux persisted in lipidation-deficient cells (that is, cells lacking ATG7, ATG10, or ATG3) and was sensitive to BafA1 treatment (Fig 1C) or sgATG9A (Fig EV1B). Consistent with these data, fluorescence-based microscopy of tf-NBR1-expressing cell lines revealed diffuse, RFP-only signal in wild-type and lipidation-deficient cell lines, while large RFP$^+$/GFP$^+$ puncta were apparent in $ATG9A^{KO}$ and $FIP200^{KO}$ cells (Figs 1D and EV1C). Acute (8 h) expression of BFP-tagged FIP200 in $FIP200^{KO}$ cells rescued flux of both tf-NBR1 aggregates and soluble tf-LC3 with comparable efficiency, suggesting that aggregated tf-NBR1 persists in a largely autophagy-competent state (Fig EV1D). Finally, correlative light and electron microscopy (CLEM) revealed the association of tf-NBR1 with double-membrane vesicles in $ATG7^{KO}$ cells (Figs 1E and EV1E). Collectively, these data are indicative of a basal autophagic flux that persists in the absence of lipidated LC3.

Our data indicate that autophagy receptors are differentially able to engage in lipidation-independent autophagy with NDP52 ≪SQSTM1 < TAX1BP1 < NBR1 under these conditions (Fig 1B). ATG7-independent autophagosomes have been reported to be smaller and form less efficiently than in wild-type cells (Nguyen *et al*, 2016; Tsuboyama *et al*, 2016; Vaites *et al*, 2017). However, in our system, NBR1 turnover persists with comparable efficiency. In an attempt to unify these observations, we considered whether robustness—defined here as the ability to efficiently degrade cargo in response to increasing burden—was influenced by LC3 lipidation. To explore this possibility, we transiently transfected wild-type and knockout cell lines with tf-NBR1 driven from a CMV promoter. The

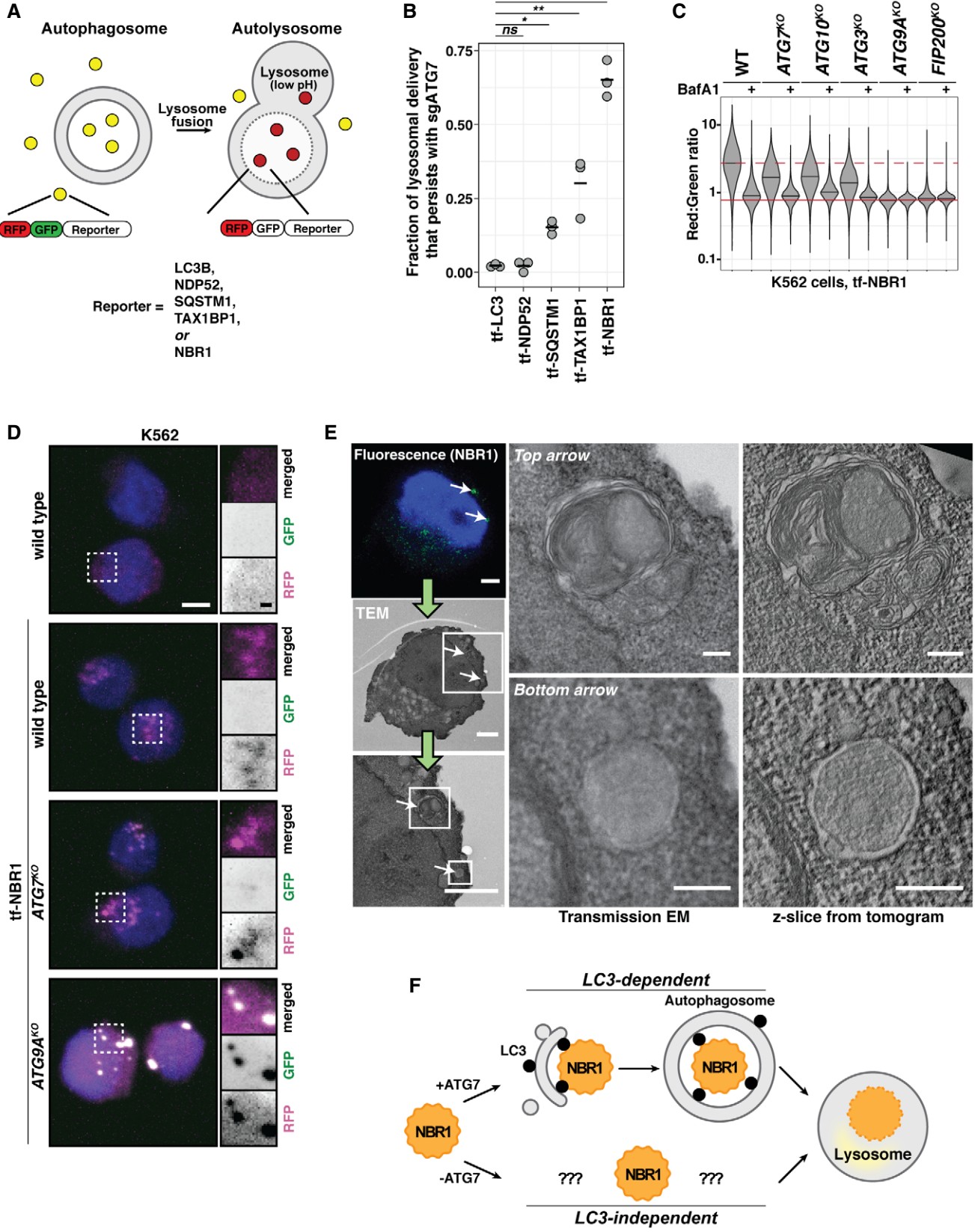

**Figure 1.**

◀

**Figure 1.   Lysosomal degradation of NBR1 persists in an ATG7-independent manner.**

A   Schematic depicting the tf-Reporter system. Both RFP and GFP fluorescence are observed throughout autophagosome formation. Upon autolysosome formation, GFP fluorescence is quenched due to the low pH of the lysosomal lumen. Corresponding changes in red:green ratio are reflective of reporter delivery to autolysosomes.

B   Dot plot indicating fraction of tf-Reporters delivered to the lysosome by ATG7-independent mechanisms. The median red:green ratio of each population was used to calculate fractional delivery according to the following formula: $(ratio_{sgATG7}-ratio_{sgATG9A})/(ratio_{NegativeCtl1}-ratio_{sgATG9A})$, with the assumption that sgATG9A yields a true autophagy-null phenotype. Mean values for each reporter are indicated by a black bar. $n = 3$ for each reporter. $P$ values were determined using a one-way ANOVA ($P < 0.0001$) with Dunnett's multiple comparison test to compare each sample to the tf-LC3 control. *$P < 0.05$; **$P < 0.005$; ***$P < 0.001$; ns, not significant.

C   Wild-type (WT) K562 cells and indicated deletion isolates are treated with 100 nM Bafilomycin A1 (BafA1) or DMSO for 18h and analyzed by flow cytometry for red:green ratio of tf-NBR1. Median values for each sample are identified by a black line within each violin. The red dotted line across all samples corresponds to the red:green ratio in wild-type cells. The red solid line across all samples corresponds to the ratio observed under maximally inhibited conditions ($ATG9A^{KO}$ cells). $n = 10,000$ cells. See Fig EV1A for validation of genotype.

D   Representative confocal micrographs (as maximum intensity projections) of wild-type K562 cells and indicated deletion isolates expressing tf-NBR1. Selected regions (white box) of micrographs are shown as single and merged channels from fluorescence microscopy. Red fluorescent protein (RFP), magenta; Green fluorescent protein (GFP), green; Hoechst, blue. Scale bars: large panels, 5 μm; small panels, 1 μm. All images scaled equally. See Fig EV1C for related images.

E   Correlative light and electron microscopy (CLEM) of K562 $ATG7^{KO}$ cells expressing tf-NBR1 under basal conditions. Analysis workflow is indicated by green arrows. White arrows indicate representative structures of interest. White boxes demarcate zoomed areas in subsequent images. NBR1, green; Hoechst, blue. Scale bar (small images), 2.5 μm. Scale bar (large images), 250 nm. See Fig EV1E for images of additional structures.

F   Presumptive model for ATG7-independent autophagy. Delivery of NBR1 to the lysosome is dependent on FIP200 and ATG9A (not shown). In the presence of LC3-lipidation, NBR1 incorporation into autophagosomes is driven by receptor interactions with LC3. In the absence of lipidated LC3, NBR1 is selectively delivered to the lysosome by a largely unknown mechanism.

Data information: See also Fig EV1.
Source data are available online for this figure.

resulting transfected cell populations contained a broad distribution of tf-NBR1 expression levels. We then assessed robustness by determining whether fractional turnover (a proxy for autophagic efficiency) scaled with autophagic burden (tf-NBR1 expression levels). In wild-type cells, fractional turnover of NBR1 was nearly complete at all expression levels (Fig EV1F, blue). In contrast, in $ATG7^{KO}$ cells (Fig EV1F, red), autophagic capacity was readily overwhelmed; as expression levels increased, the fractional turnover of tf-NBR1 dropped precipitously, approaching levels of inhibition observed in $ATG9A^{KO}$ cells (Fig EV1F, black). Therefore, lipidated LC3 can support NBR1 turnover although it is not required for autophagosome formation or NBR1 incorporation (Fig 1F).

**Genome-wide screening reveals factors required for ATG7-independent autophagy**

Since lysosomal delivery of NBR1 persists in cells lacking lipidated LC3, we reasoned that tf-NBR1 presents a genetically tractable system for deconvolving LC3-dependent and LC3-independent mechanisms of autophagy. To this end, we performed genome-wide, CRISPR-mediated, knockout screens for modulators of tf-NBR1 trafficking in three related deletion backgrounds in which LC3 lipidation is inhibited: $ATG7^{KO}$, $ATG3^{KO}$, and $ATG10^{KO}$ cells. We used the Brunello, two-vector, sgRNA library containing 76,441 sgRNAs spanning 19,114 genes (Doench $et al$, 2016). Following lentiviral integration and selection, we used fluorescence activated cell sorting (FACS) to collect the top and bottom third of cells according to red:green ratio (Appendix Fig S1A). Read counts of sgRNAs in each pool were obtained by Illumina sequencing (Table EV1) and analyzed using MAGeCK (Li $et al$, 2014, 2015). To facilitate the comparison of these results with previous studies, we employed the beta score (similar to log-fold change) as a proxy for the strength of a gene as an autophagy effector (Table EV2).

The genetic modifiers of ATG7-, ATG3-, and ATG10-independent autophagy (hereafter ATG7-independent autophagy) overlapped as anticipated for mutants in a shared pathway (Fig 2A, Table EV2). Top modifiers consisted of canonical autophagy factors including

ATG9A, FIP200, ATG101, RAB7A, PIK3C3, and ATG14 (Appendix Fig S1B). Notably absent were other components of the lipidation machinery (e.g., ATG5, ATG12, etc.) and known binding partners of LC3 (e.g., EPG5), consistent with their function in a shared pathway. To identify unique modifiers of ATG7-independent autophagy, we compared tf-NBR1 modifiers in wild-type and lipidation-deficient cells (Appendix Fig S1C) or lipidation-dependent autophagy (as represented by tf-NDP52 (Shoemaker $et al$, 2019)) and lipidation-independent autophagy (tf-NBR1 in lipidation-deficient cells (Fig 2B)). From this comparative analysis, we identified several unique modifiers of ATG7-independent autophagy (e.g., TAX1BP1, discussed below). To confirm these data, we transduced our reporter cell lines with individual sgRNAs and monitored red:green ratio by flow cytometry (Fig 2C). As expected, the effect of individual sgRNAs was largely reflective of each gene's beta score.

**TAX1BP1 is a potent effector of ATG7-independent autophagy**

Our screening approach identified five genes as highly selective modifiers of LC3-lipidation-independent autophagy: TAX1BP1, TBK1, GDI2, TRAPPC11, and KAT8. Each of these genes possessed a beta score > 0.5 in all three lipidation-deficient cell lines, but a beta score < 0.1 for canonical autophagy (i.e., tf-NDP52; Fig 2B, Table EV2). Of these, the top genetic modifier we identified for ATG7-independent autophagy was TAX1BP1. Thus, TAX1BP1 is both a substrate (Fig 1B) and a facilitator of ATG7-independent autophagy. To validate TAX1BP1 as a specific modifier of ATG7-independent flux, we generated single knockout ($ATG9A^{KO}$, $ATG7^{KO}$, $TAX1BP1^{KO}$) and double knockout ($ATG7^{KO}/TAX1BP1^{KO}$) cell lines (Fig EV2A). Bafilomycin-responsive flux of tf-NBR1 persisted in $ATG7^{KO}$ or $TAX1BP1^{KO}$ cells, however flux was fully inhibited in $ATG7^{KO}/TAX1BP1^{KO}$ cells (Fig 3A). Consistent with these findings, tf-NBR1 coalesced into large RFP$^+$/GFP$^+$ puncta in $ATG7^{KO}/TAX1BP1^{KO}$ cells, similar to other autophagy-null alleles (Fig 3B, compare with Fig 1D). CLEM visualization of tf-NBR1 puncta in $ATG7^{KO}/TAX1BP1^{KO}$ cells revealed an electron-dense structure similar to recently reported cytosolic p62/SQSTM1 bodies

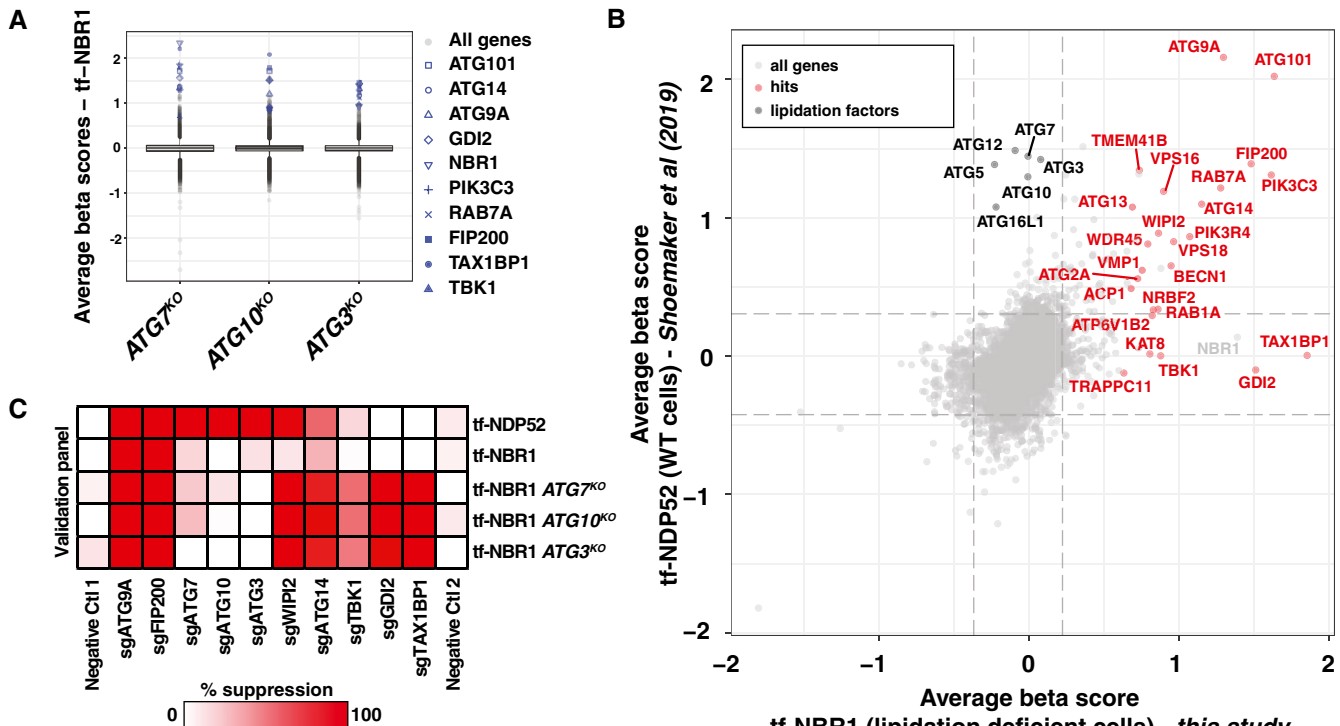

**Figure 2. Identification of genes required for ATG7-independent autophagy.**

A Box plot of average beta scores (similar to log-fold change) for tf-NBR1 flux in indicated deletion cell lines. Median values (central band), inner quartiles (boxed regions), 1.5 interquartile ranges (whiskers), and outliers (dots) are indicated. Representative hits across all three genetic backgrounds are indicated in blue.

B Gene correlation plot of average beta scores for lipidation-dependent autophagy (tf-NDP52, from (Shoemaker *et al*, 2019)) and lipidation-independent autophagy (average of tf-NBR1 from *ATG7^KO*, *ATG10^KO*, and *ATG3^KO* cells, Table EV2). Highlighted in red are genes with a beta score > 0.5 across all three lipidation-deficient cell lines. In black are lipidation components. Dashed lines, top 1% of beta scores.

C Wild-type K562 cells and indicated deletion isolates expressing Cas9 and tf-NDP52 or tf-NBR1 were transduced with individual sgRNAs for indicated genes. The median red:green ratio of each population was used to calculate the fold-repression according to the following formula: (ratio_sgGene−ratio_sgATG9A)/(ratio_NegativeCtl−ratio_sgATG9A), with the assumption that sgATG9A yields a true autophagy-null phenotype. Deeper shades of red indicate stronger suppressor phenotypes. Genes were clustered on the basis of their patterns of genetic interactions.

Data information: See also Appendix Fig S1.
Source data are available online for this figure.

(Jakobi *et al*, 2020) and suggested the absence of an enclosing membrane (Fig 3C).

Expression of either TAX1BP1 or ATG7 was sufficient to rescue tf-NBR1 flux in *ATG7^KO/TAX1BP1^KO* cells, confirming the specificity of the deletions and the sufficiency of either factor to enable NBR1 flux (Fig EV2B and C). In contrast, a catalytically dead variant of ATG7 (ATG7^C572S) did not rescue (Fig EV2C). To fully distinguish LC3 lipidation from ATG7 activity, we monitored autophagic flux in the presence of RavZ, an effector protein from *Legionella pneumophila* that irreversibly cleaves lipidated Atg8 homologs (Choy *et al*, 2012). This approach allows us to disrupt LC3 lipidation without altering the activity of the lipidation machinery. We used a BFP-tagged RavZ construct to distinguish transfected cells and monitored autophagy by red:green ratio using flow cytometry. To validate the approach, we transfected RavZ into cells expressing tf-LC3 or tf-NDP52. As predicted, RavZ expression greatly reduced the lysosomal delivery of both reporters (75 and 80% inhibited, respectively, Fig EV2D). In comparison, RavZ had only a modest inhibitory effect on tf-NBR1 (28% inhibited, Fig 4A). However, in combination with *TAX1BP1* deletion, RavZ had a significantly larger effect on tf-NBR1

flux (63% inhibited in *TAX1BP1^KO* vs 28% in wild type). As a control, RavZ had no effect on NBR1 flux in *ATG7^KO* cells, where lipidation is already compromised. Collectively, these data substantiate the interpretation that TAX1BP1 facilitates lipidation-deficient autophagy.

When weighing alternative hypotheses, we considered a recent report that SQSTM1 can recruit soluble LC3B (LC3-I), especially when autophagy is inhibited (Runwal *et al*, 2019). To evaluate whether NBR1 puncta recruit soluble LC3 and/or whether LC3-I might contribute to trafficking (e.g., as in EDEMosomes (Calì *et al*, 2008)), we used immunofluorescence microscopy to assess colocalization of LC3A/B and tf-NBR1 in various deletion cell lines. However, we were unable to find LC3 puncta that colocalized with tf-NBR1 in any lipidation-deficient cell line, suggesting that LC3-I is unlikely to play a role in the ATG7-independent trafficking of tf-NBR1 (Fig EV2E).

To ensure that ATG7-independent autophagy was not an artifact of the tf-Reporter system (e.g., due to tf-NBR1 over-expression, and tagging artifacts), we generated a panel of deletions in cells lacking tf-NBR1 and monitored endogenous receptor levels (Figs 4B and

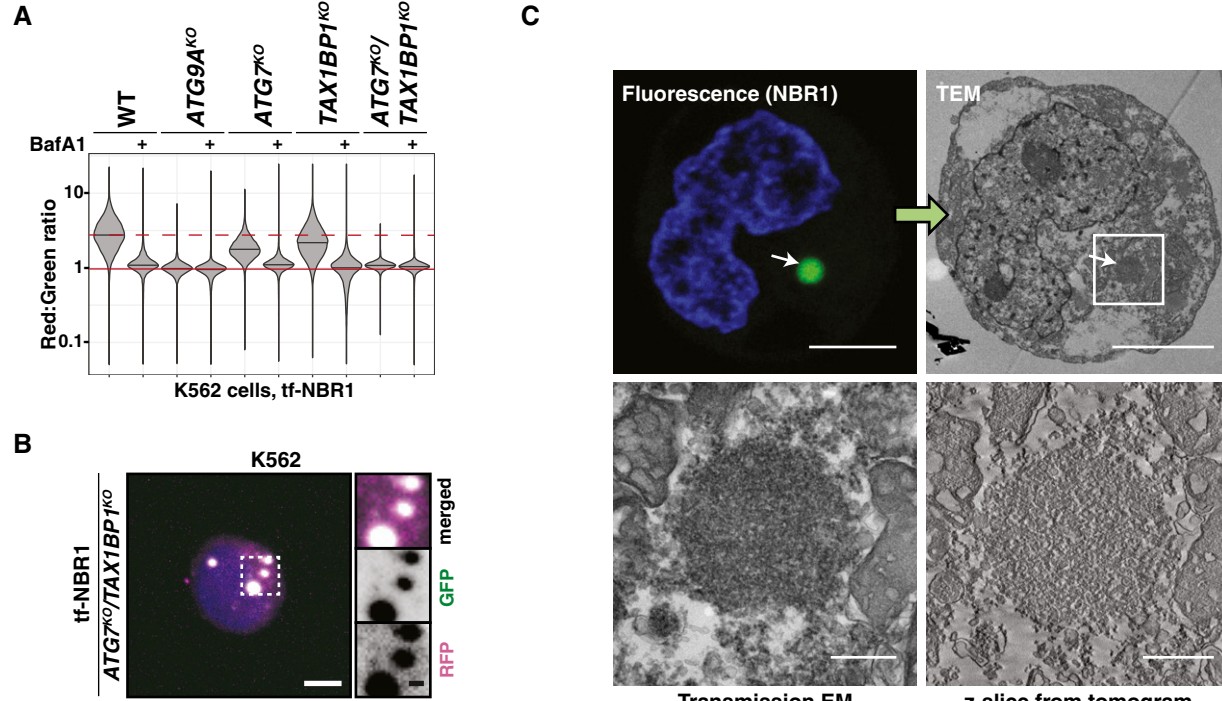

**Figure 3. tf-NBR1 flux requires TAX1BP1 in the absence of ATG7.**

A   Wild-type (WT) K562 cells and indicated deletion isolates were treated with 100 nM Bafilomycin A1 (BafA1) or DMSO for 18h and analyzed by flow cytometry for red:green ratio of tf-NBR1. Median values for each sample are identified by a black line within each violin. The red dotted line across all samples corresponds to the red:green ratio in wild-type cells. The red solid line across all samples corresponds to the ratio observed under maximally inhibited conditions ($ATG9A^{KO}$ cells). ($n > 10,000$ cells). See Fig EV2A for validation of genotype.

B   Representative confocal micrograph (as maximum intensity projections) of $ATG7^{KO}/TAX1BP1^{KO}$ K562 cells expressing tf-NBR1. Selected region (white box) of micrograph is shown as single and merged channels from fluorescence microscopy. RFP, magenta; GFP, green; merged, white; Hoechst, blue. Scale bars: large panel, 5 μm; small panels, 1 μm. Compare to Figs 1D and EV1C.

C   Correlative light and electron microscopy (CLEM) of K562 $ATG7^{KO}/TAX1BP1^{KO}$ cells expressing tf-NBR1 under basal conditions. Analysis workflow is indicated by green arrows. White arrow indicates a structure of interest. White box demarcates zoomed area in bottom images. NBR1, green; Hoechst, blue. Scale bar: top panels, 5 μm; bottom panels, 500 nm.

Source data are available online for this figure.

EV2F). As expected, endogenous NBR1 levels responded differently to deletion of *ATG7* and *ATG9A*, showing greater accumulation in $ATG9A^{KO}$ cells. Furthermore, in the double knockout ($ATG7^{KO}/TAX1BP1^{KO}$), NBR1 levels rose above those observed from either *ATG7* or *TAX1BP1* deletion alone. Quantification of mRNA levels by qRT–PCR confirmed that the observed changes in NBR1 protein were not due to transcriptional effects (Fig EV2G). Meanwhile, levels of endogenous NDP52 and SQSTM1 increased similarly in $ATG9A^{KO}$ and $ATG7^{KO}$ cell lines and the double deletion had no synergistic effect on NDP52 or SQSTM1. NBR1 and TAX1BP1 were also refractory to *ATG7* deletion in HEK293T cells when compared to $ATG9A^{KO}$ cells or BafA1-treated controls (Fig EV2H), while NDP52 and SQSTM1 were inhibited across all conditions. To confirm that differences in receptor levels between knockout cell lines reflected differences in autophagic flux, we performed an established protease protection assay to monitor vesicle formation (Fig 4C). Wild-type and ATG-deficient K562 cells were treated with 50 nM Bafilomycin A1 to allow autophagic vesicles to accumulate. Cells were then lysed by mechanical disruption, subjected to trypsin-mediated proteolysis and analyzed by immunoblotting

(Figs 4D and E, and EV2I). Wild-type cells and $ATG9A^{KO}$ cells were used as positive and negative controls, respectively. As expected, all receptors were protected from proteolysis in wild-type cells and sensitive to proteolysis in $ATG9A^{KO}$ cells (Fig 4D and E). In $ATG7^{KO}$ cells, receptors showed differential effects: NBR1 and TAX1BP1 were protected while NDP52 was sensitive to proteolysis. However, NBR1 became sensitive to proteolysis in $ATG7^{KO}/TAX1BP1^{KO}$ cells. Taken together, these data (i) reaffirm the differential effect of $ATG7^{KO}$ and $ATG9A^{KO}$ on autophagy receptors, (ii) verify TAX1BP1 as both a substrate of ATG7-independent autophagy and a facilitator of ATG7-independent flux, and (iii) indicate that the autophagy defect in $ATG7^{KO}/TAX1BP1^{KO}$ cells is due to an inability to sequester NBR1 in a membrane compartment.

**NBR1 forms a heterotypic receptor complex that requires TAX1BP1 to induce local autophagosome formation**

Concomitant turnover of TAX1BP1 and NBR1 suggests the formation of a heterotypic receptor complex. To test this, we immunoprecipitated tf-NBR1 from detergent-solubilized extracts and analyzed

the immunoprecipitate by Western blot. In the background of multiple ATG deletions, we observed an association between NBR1, TAX1BP1, and SQSTM1 (Fig 5A). The association of NBR1, TAX1BP1, and SQSTM1 persisted in the absence of ATG9A or

FIP200 indicating that co-recruitment of receptors to early autophagosomal membranes is not a precondition for receptor association. In a reciprocal experiment, tf-TAX1BP1 similarly co-immunoprecipitated endogenous NBR1 and SQSTM1 in wild-type

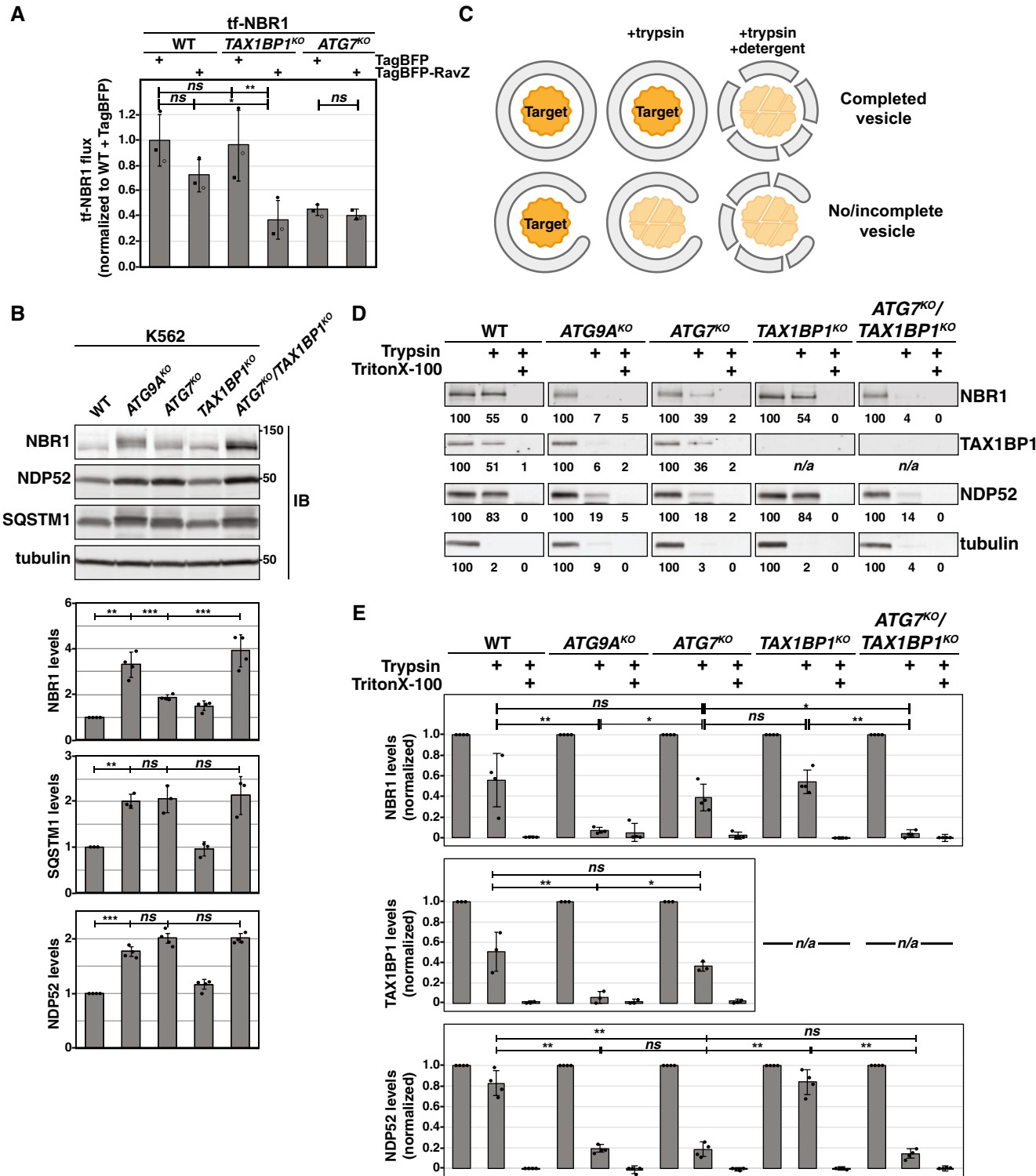

Figure 4.

**Figure 4. TAX1BP1 is required for lysosomal degradation of NBR1 in the absence of LC3 lipidation.**

A Wild-type and indicated K562 knockout cells expressing tf-NBR1 were transfected with TagBFP or TagBFP-RavZ, and BFP-positive cells were analyzed for red:green ratio by flow cytometry. The median red:green ratio of each sample was used to calculate flux relative to WT + TagBFP. Bar graphs represent mean ± SD from three independent experiments. $n > 7,500$ cells per sample. $P$ values were determined using a repeated-measures one-way ANOVA ($P = 0.0001$) with Tukey's HSD post-test. **$P < 0.005$; *$P < 0.05$; ns, not significant. See also Fig EV2D.

B K562-derived extracts prepared from wild-type (WT) and clonal deletion isolates were resolved by SDS–PAGE followed by immunoblotting (IB) with indicated antibodies. All samples were normalized by total protein using a BCA assay prior to loading. Bar graphs represent mean ± SD from independent experiments ($n = 4$ {NBR1, NDP52}; $n = 3$ {SQSTM1}). $P$ values for WT vs $ATG9A^{KO}$ were determined using a one sample test (theoretical mean = 1) with Bonferroni correction. All other $P$ values were determined using a one-way ANOVA ($P < 0.001$) with Tukey's HSD post-test. ***$P < 0.001$, **$P < 0.01$; ns, not significant.

C Schematized representation of protease protection assay for detecting vesicle closure. Cells were treated for 18 h with 50 nM Bafilomycin A1 prior to mechanical lysis. The corresponding cell extracts were treated as indicated prior to being resolved by SDS–PAGE and analyzed by immunoblotting.

D Wild-type and indicated K562 knockout cells were treated as described in (C). The corresponding cell extracts were resolved by SDS–PAGE and analyzed by immunoblotting (IB) with indicated antibodies. Shown are representative images from one experiment with mean intensity levels from (E) indicated below each image (NBR1: $n = 4$; TAX1BP1: $n = 3$, NDP52: $n = 4$, tubulin: $n = 4$).

E Quantitation of protease protection data from experiments in (D). Bar graphs show the mean ± SD of each sample from independent experiments. Trypsin-treated samples were compared using a one-way ANOVA (NBR1: $n = 4$, $P = 0.0001$; TAX1BP1: $n = 3$, $P = 0.0098$; NDP52: $n = 4$, $P < 0.0001$) with Tukey's HSD post-test. **$P < 0.01$; *$P < 0.05$; ns, not significant.

Data information: See also Fig EV2I.
Source data are available online for this figure.

cells (Fig EV3A). In contrast, NDP52 was absent from this complex of receptors. This suggests that NDP52 is subject to a different regulatory module and may provide a rationale for why NDP52 does not undergo ATG7-independent autophagy under these experimental conditions. Quantitative mass spectrometry analysis of the tf-NBR1 co-immunoprecipitate confirmed TAX1BP1 and SQSTM1 as top interactors of tf-NBR1 by fold-enrichment (Table EV3). One additional factor, TROVE2, was comparably enriched. However, while we confirmed the association of NBR1 and TROVE2 by immunoblotting (Fig EV3B), we note that total cellular levels of TROVE2 were not influenced by autophagy (Fig EV3C).

To further analyze the interplay between autophagy receptors, we used immunofluorescence microscopy to monitor the spatial dynamics of receptors *in vivo*. In response to autophagy inhibition (e.g., in $ATG9A^{KO}$ cells), tf-NBR1 accumulated *in vivo* and coalesced into puncta that colocalized with TAX1BP1 and SQSTM1 (Fig 5B). In contrast, NDP52 did not associate with tf-NBR1. Rather, NDP52 remained diffuse in the cytosol (Fig 5B). The differential behavior of TAX1BP1 and NDP52 was independently observed when we deleted $ATG9A$ from cells expressing tf-TAX1BP1 or tf-NDP52. Upon transduction of sgATG9A constructs, lysosomal delivery of both reporters was fully inhibited (Fig EV3D and E), but tf-TAX1BP1 formed punctate structures while tf-NDP52 remained diffuse (Fig 5C). Collectively, these data are consistent with the coalescing of NBR1, TAX1BP1, and SQSTM1 into a heterotypic receptor complex that defines an ATG7-independent autophagy substrate.

## TAX1BP1 function in ATG7-independent autophagy is independent of its ubiquitin-binding zinc finger (UBZ) domains and requires a novel recruitment modality

Within the heterotypic receptor complex, what role does each receptor play? SQSTM1 and NBR1 are known to interact directly through their PB1 domains, and multiple studies have documented the functional relationship between NBR1 and SQSTM1 (Zaffagnini *et al*, 2018; Jakobi *et al*, 2020; Sánchez-Martín *et al*, 2020). Since ectopic tf-NBR1 expression increases total NBR1 expression levels (Fig 5A), this may influence receptor dynamics. To evaluate the interplay

between SQSTM1 and NBR1, we first transduced tf-NBR1 cell lines with sgSQSTM1 or a non-targeting control. Knockout efficiency was monitored by immunoblotting (Fig EV3F). Consistent with previous results (Kirkin *et al*, 2009), NBR1 flux was not affected by deletion of SQSTM1 (Fig EV3G). Conversely, overexpression SQSTM1 also had no effect NBR1 flux (Fig EV3H). Likewise, NBR1 overexpression, which was shown recently to inhibit SQSTM1 flux in another system, did not impact tf-SQSTM1 flux under our experimental setup (Fig EV3I) (Sánchez-Martín *et al*, 2020). Together, our data suggest that the SQSTM1-NBR1 interaction does not play a critical role in receptor flux under these conditions.

Next, to distinguish the various functions of TAX1BP1, we generated N- and C-terminal truncation mutants (Fig 6A). We then compared the ability of these variants to support ATG7-independent autophagy or be targeted by canonical autophagy. To assess ATG7-independent autophagy, we transfected $ATG7^{KO}/TAX1BP1^{KO}$ cells with BFP-tagged TAX1BP1 variants and assessed the ability of each construct to restore tf-NBR1 flux (Fig 6B, Appendix Fig S2A and B). For canonical autophagy, we monitored the lysosomal delivery of tf-TAX1BP1 variants in $TAX1BP1^{KO}$ cells (Fig 6B, Appendix Fig S2C and D). Through this structure-function approach, we found multiple regions that differentiate TAX1BP1's function in ATG7-independent autophagy from its role as a canonical autophagy substrate. This includes the UBZ domains, coiled-coil 2, and the SKICH domain (discussed below).

TAX1BP1 contains an N-terminal SKICH domain, three coiled-coil (CC) regions, and two C-terminal UBZ domains (Fig 6A). All C-terminal truncations of TAX1BP1, which eliminate the UBZ domains, were deficient as canonical autophagy substrates (TAX1BP1$^{1–724}$—26% vs WT) (Fig 6B, Appendix Fig S2C). In contrast, C-terminal truncations up to and including TAX1BP1$^{1–506}$ fully restored NBR1 flux in $ATG7^{KO}/TAX1BP1^{KO}$ cells (TAX1BP1$^{1–506}$—182% vs WT) (Fig 6B, Appendix Fig S2A). Thus, coiled-coil 3 (CC3) and the UBZ domains of TAX1BP1 are fully dispensable for ATG7-independent autophagy. Further C-terminal truncations (past TAX1BP1$^{1–506}$) failed to rescue NBR1 flux in $ATG7^{KO}/TAX1BP1^{KO}$ cells indicating that coiled-coil 2 (CC2) plays a critical role in ATG7-independent autophagy. CC2 overlaps two sequences previously implicated in the self-oligomerization of TAX1BP1, TAX1BP1$^{320–420}$

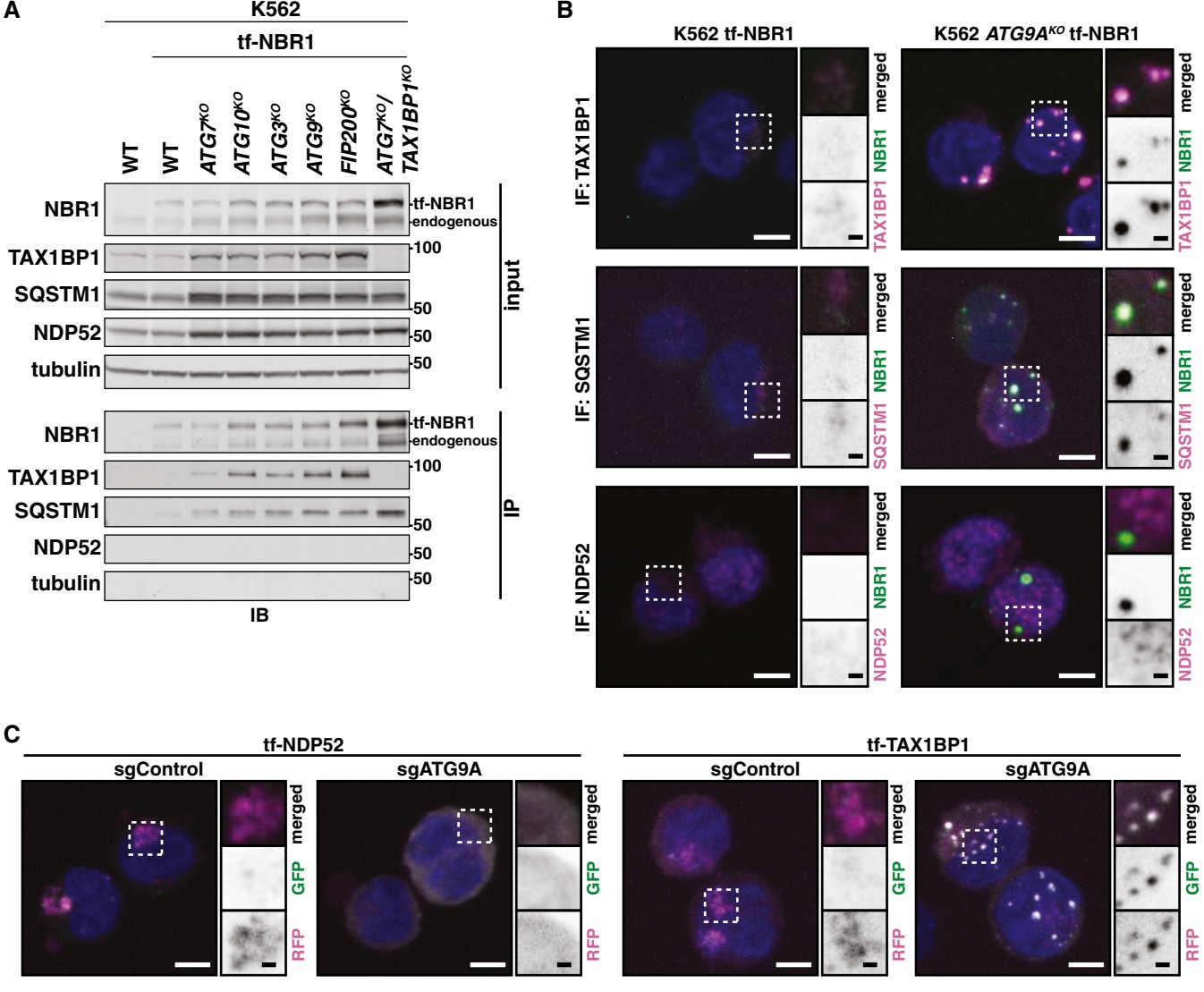

**Figure 5. A heterotypic receptor complex defines an ATG7-independent substrate.**

A   Extracts derived from wild-type and deletion K562 cells expressing tf-NBR1 were normalized for total protein by BCA. NBR1 was immunoprecipitated using GFP-Trap dynabeads. Input and eluate were resolved by SDS–PAGE followed by immunoblotting (IB) with indicated antibodies.

B   Representative confocal micrographs (as maximum intensity projections) of wild-type K562 and *ATG9A^KO* cells expressing tf-NBR1. Selected regions (white box) of micrographs are shown as single and merged channels from fluorescence and immunofluorescence microscopy against indicated proteins. In conditions where puncta were not observed, representative cytoplasmic regions were selected to showcase diffuseness of signal. TAX1BP1, SQSTM1, or NDP52, magenta; NBR1, green; Hoechst, blue. Scale bars: large panels, 5 μm; small panels, 1 μm.

C   Shown are representative confocal micrographs (as maximum intensity projections) of K562 cells expressing tf-NDP52 (or tf-TAX1BP1) and transduced with sgRNAs targeting ATG9A (sgATG9A) or a negative control (sgControl). Selected regions (white box) of micrographs are shown as single and merged channels from fluorescence microscopy. RFP, magenta; GFP, green; Hoechst, blue. Scale bars: large panels, 5 μm; small panels, 1 μm. See Fig EV3D and E for confirmation of ATG9A deletion.

Data information: See also Fig EV3.
Source data are available online for this figure.

(the O domain (Ling & Goeddel, 2000), and TAX1BP1^446–600 (Chin *et al*, 2007). To confirm a role for this region in self-oligomerization, we performed co-immunoprecipitation experiments between myc-TAX1BP1^WT and BFP-V5-tagged TAX1BP1 truncations (Fig 6C). Consistent with previous reports, truncations that disrupted CC2 inhibited the ability of TAX1BP1 to oligomerize. Moreover, internal deletion of the O domain was sufficient to disrupt oligomerization of the TAX1BP1^1–506 fragment, although it was not sufficient in the

context of full length TAX1BP1 (Fig EV4A). We conclude that multiple regions within TAX1BP1, including but not limited to the O domain, facilitate self-oligomerization.

While mutations within CC2 disrupted oligomerization, we note that oligomerization did not strictly correlate with the ability of TAX1BP1 variants to complement *ATG7^KO/TAX1BP1^KO* cells. Specifically, TAX1BP1^1–506ΔO domain did not oligomerize but generally supported ATG7-independent autophagy (65% vs WT). This

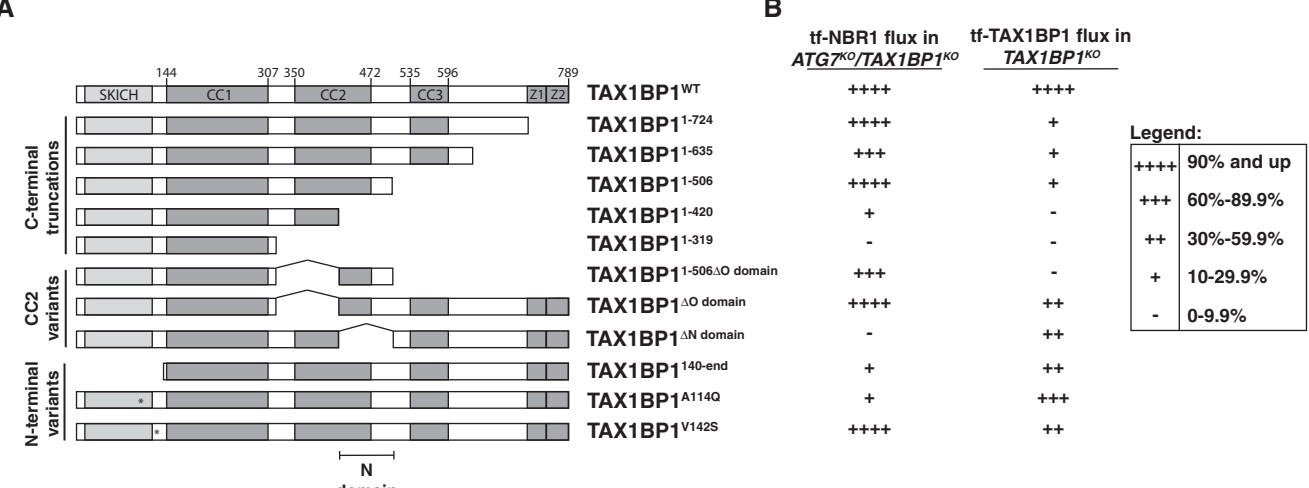

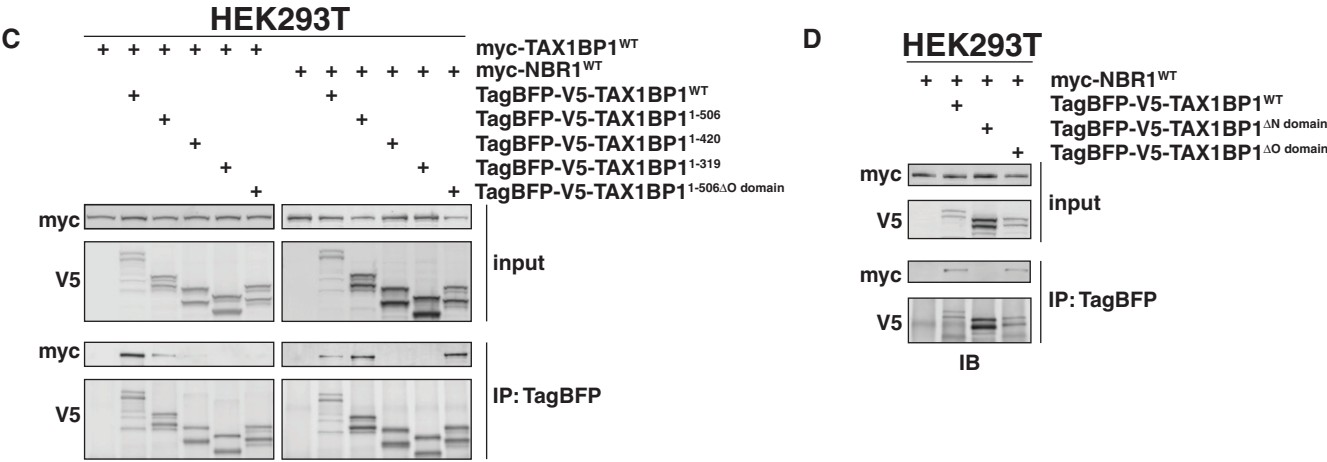

**Figure 6. Domain analysis of TAX1BP1 function distinguishes canonical and non-canonical autophagy functions.**

A   Domain organization of TAX1BP1 (NP_006015.4) and derived variants. Coiled-coil regions were defined by PCOILS using a 28 amino acid window and cutoff of 0.5. SKICH, SKIP carboxyl homology domain; CC, coiled-coil; Z1 and Z2, ubiquitin-binding zinc finger (UBZ) domains; N, NBR-interacting domain.

B   A summary of TAX1BP1 variants' activity in complementation and flux analyses. For complementation analysis, $ATG7^{KO}/TAX1BP1^{KO}$ K562 cells expressing tf-NBR1 were electroporated with BFP-TAX1BP1 constructs, and BFP-positive cells were analyzed for red:green ratio (for underlying data, see Appendix Fig S2A and B). For flux analysis, tf-TAX1BP1 variants were transfected into $TAX1BP1^{KO}$ K562 cells and assessed for red:green ratio by flow cytometry (for underlying data, see Appendix Fig S2C and D). In all experiments, median red:green ratios were used to compare populations.

C   HEK293T cells were transfected with either myc-TAX1BP1$^{WT}$ or myc-NBR1$^{WT}$ and indicated TagBFP-V5-TAX1BP1 variants. Extracts derived from transfected cells were immunoprecipitated (IP) with anti-TagBFP magnetic beads. Input and eluates were resolved by SDS–PAGE followed by immunoblotting (IB) with indicated antibodies.

D   HEK293T cells were transfected with myc-NBR1$^{WT}$ and indicated TagBFP-V5-TAX1BP1 variants. Extracts derived from transfected cells were immunoprecipitated (IP) with anti-TagBFP magnetic beads. Input and eluates were resolved by SDS–PAGE followed by immunoblotting (IB) with indicated antibodies.

Data information: See also Fig EV4G and H.
Source data are available online for this figure.

observation suggested that the second half of CC2 (TAX1BP1$^{420–506}$) is required for ATG7-independent autophagy for reasons other than self-oligomerization. As such, we considered other critical functions that CC2 might perform. We note that NBR1 puncta colocalize with ubiquitin (Fig EV4B), and the ubiquitin-binding (UBA) domain of NBR1 strongly influences flux (Fig EV4C). This contrasts with TAX1BP1, where the ubiquitin-binding domains are dispensable for ATG7-independent autophagy (Fig 6B). Thus, it was unclear how

TAX1BP1 and NBR1 might be coordinately recruited to puncta. We hypothesized that TAX1BP1$^{420–506}$ directly facilitates an interaction between TAX1BP1 with NBR1. To test this, we co-transfected myc-tagged NBR1 with BFP-V5-tagged TAX1BP1 variants, immunoprecipitated TAX1BP1 and assessed co-immunoprecipitation of NBR1 by immunoblotting (Fig 6C). Strikingly, the second half of CC2 (TAX1BP1$^{420–506}$) was fully required for TAX1BP1 to associate with NBR1 while the first half of CC2 (the O domain) had no effect on

NBR1 binding (Fig 6C and D). The effect of the N domain was confirmed using fluorescence microscopy (Fig EV4D). Using $ATG9A^{KO}$/$TAX1BP1^{KO}$ cells, we transduced BFP-tagged TAX1BP1 variants and assessed colocalization with tf-NBR1 puncta. As expected, BFP-TAX1BP1$^{1-506}$ coalesced into puncta that colocalized with NBR1, while BFP-TAX1BP1$^{1-420}$ remained diffuse. Thus, the second half of CC2 (hereafter the NBR1-binding domain, or N domain) defines a ubiquitin-independent mode of TAX1BP1 recruitment to NBR1 that is critical for NBR1 flux in cells lacking ATG7.

### TAX1BP1 recruits FIP200 to NBR1 puncta to induce local autophagosome formation

N-terminal truncations of TAX1BP1 also failed to rescue $ATG7^{KO}$/$TAX1BP1^{KO}$ cells, indicating the N-terminal SKICH domain is required for ATG7-independent autophagy (Fig 6B). The SKICH domain of TAX1BP1 is known to bind FIP200 and TBK1 (Fu *et al*, 2018; Ravenhill *et al*, 2019), both of which we identified as critical for NBR1 flux in $ATG7^{KO}$ cells (Figs 2C and EV4E and F). Mutation of Ala$^{119}$→Gln within the NDP52 SKICH domain was recently reported to disrupt the interaction of NDP52 and TBK1 (Ravenhill *et al*, 2019). To assess the effect of the analogous mutation (Ala$^{114}$→Gln) in TAX1BP1, we transfected HEK293T cells with V5-tagged TAX1BP1 variants and tested the ability to co-immunoprecipitate FIP200 and TBK1. As predicted, TAX1BP1$^{A114Q}$ was deficient in binding TBK1 (Fig EV4G). However, TAX1BP1$^{A114Q}$ also failed to bind FIP200, which was not observed for the analogous mutation in NDP52 (Fig EV4H) (Ravenhill *et al*, 2019). When transfected into $ATG7^{KO}$/$TAX1BP1^{KO}$ cells, TAX1BP1$^{A114Q}$ was deficient in ATG7-independent autophagy (29% compared to WT) (Fig 6B, Appendix Fig S2A). However, tf-TAX1BP1$^{A114Q}$ exhibited minimal defect for canonical autophagy in $TAX1BP1^{KO}$ cells (74% compared to WT; Fig 6B, Appendix Fig S2C). This represents a striking ability to separate the function of TAX1BP1 in these processes, dependent solely on TAX1BP1's ability to bind TBK1 and/or FIP200.

The above data suggest TAX1BP1 is the linchpin that directs local autophagosome formation around NBR1 puncta through the recruitment and clustering of FIP200 and/or TBK1. To visualize early autophagosome formation processes *in vivo*, we used immunofluorescence microscopy and assessed recruitment of FIP200 and TBK1 to NBR1 puncta (Fig 7A and B). Large NBR1 puncta were found in $ATG9A^{KO}$ cells and $ATG7^{KO}$/$TAX1BP1^{KO}$ cells and, in both cases, NBR1 colocalized with TBK1 (Fig 7A). Thus, while our mutational analysis suggests that the interaction between TBK1 and TAX1BP1 is required for proper autophagosome formation, this interaction is not required for the *in vivo* recruitment of TBK1 to NBR1 puncta. In contrast, NBR1 colocalized with FIP200 in $ATG9A^{KO}$ cells but did not colocalize in $ATG7^{KO}$/$TAX1BP1^{KO}$ cells (Fig 7B). $ATG7^{KO}$/$TBK1^{KO}$ cells showed a similar phenotype to $ATG7^{KO}$/$TAX1BP1^{KO}$ cells (that is, NBR1-positive/FIP200-negative puncta), consistent with TBK1 and TAX1BP1 performing related functions (Fig EV5A).

Based on the differential recruitment of FIP200 to NBR1 in $ATG9A^{KO}$ and $ATG7^{KO}$/$TAX1BP1^{KO}$ cells, we performed an epistasis analysis by transducing wild-type, $TAX1BP1^{KO}$, and $ATG7^{KO}$/$TAX1BP1^{KO}$ cells with sgATG9A. After 8 days of puromycin selection, we analyzed colocalization of NBR1 and FIP200 using immunofluorescence microscopy (Fig 7C). Upon *ATG9A* deletion (Fig EV5B), similar numbers of NBR1 puncta were observed

regardless of the presence or absence of TAX1BP1 (1.47 vs 1.10–1.88 per cell, not significant) (Fig EV5C). However, in the absence of TAX1BP1, we observed a significant decrease in the number and percentage of NBR1 puncta that colocalized with FIP200 (92.5% vs 25–46%, $P < 0.0001$) (Figs 7D and EV5D). Furthermore, when present, the intensity of FIP200 at NBR1 puncta was also decreased (0.16 vs 0.06–0.09, $P < 0.0001$; Fig 7E). These data support that (i) TAX1BP1 bridges NBR1 and FIP200 to enforce local autophagosome formation and continued cargo specificity and (ii) this process does not require lipidated LC3 (Fig 7F).

## Discussion

Lipidated LC3 is implicated in many steps of autophagosome formation including cargo selection, membrane expansion, autophagosome trafficking, and lysosomal degradation (for review, see (Mizushima, 2020)). Yet, the relative contribution of LC3 to these different facets of autophagy can be difficult to deconvolve. This is, in part, because LC3 is both a component of autophagosome biogenesis and a substrate of autophagy. That is to say, when LC3 dynamics are disrupted (e.g., upon *ATG7* deletion) autophagosomes are perturbed, but the ability to monitor autophagosome dynamics is also compromised. With alternative approaches, studies have begun to challenge fundamental assumptions about the role of LC3 in autophagy. However, the limited throughput of many LC3-independent assays has made systematic dissection of ATG7-independent autophagy elusive.

Recently, we developed a panel of reporters that rely on tandem-fluorescent (tf) epitope tagging of mammalian autophagy receptors to measure the autophagic flux (Shoemaker *et al*, 2019). Here, we report several of those receptors whose flux is variably dependent on LC3 lipidation. This includes NBR1 flux, which is minimally perturbed by deletion of *ATG7*. We exploited this discovery to perform genome-wide CRISPR screens for modulators of autophagy in cells lacking the LC3 lipidation machinery. At the outset, it was not fully clear to what extent ATG7-independent autophagy would be macroautophagy-like. For example, upon nutrient depletion, several autophagy receptors can be degraded in an ESCRT-dependent manner (Goodwin *et al*, 2017; Mejlvang *et al*, 2018). Here, we focused exclusively on basal autophagy under nutrient-replete conditions so as to not conflate these processes. Indeed, under basal conditions, ATG7-independent autophagy remained fully dependent on early (e.g., ATG9A, FIP200) and late (HOPS) modulators of autophagy. Furthermore, ATG7-independent autophagy was more, rather than less, dependent on other classical autophagy factors (e.g., WIPI2 and ATG14) when compared to wild-type cells. This genetic interaction suggests that ATG7 (and possibly also WIPI2 and/or ATG14) are not essential for autophagosome formation *per se* but contribute to the robustness of the pathway.

The number of factors that were required specifically for ATG7-independent autophagy were limited, but informative. We identified five factors required for NBR1 flux in lipidation-deficient cell lines but not required for canonical autophagy: TAX1BP1, TBK1, GDI2, TRAPPC11, and KAT8. Within this set, TAX1BP1 was the largest modifier of flux that we observed. We also identified TBK1, a known modifier of TAX1BP1 activity, which modulated NBR1 flux similarly to TAX1BP1. As a receptor, TAX1BP1 is both a substrate of

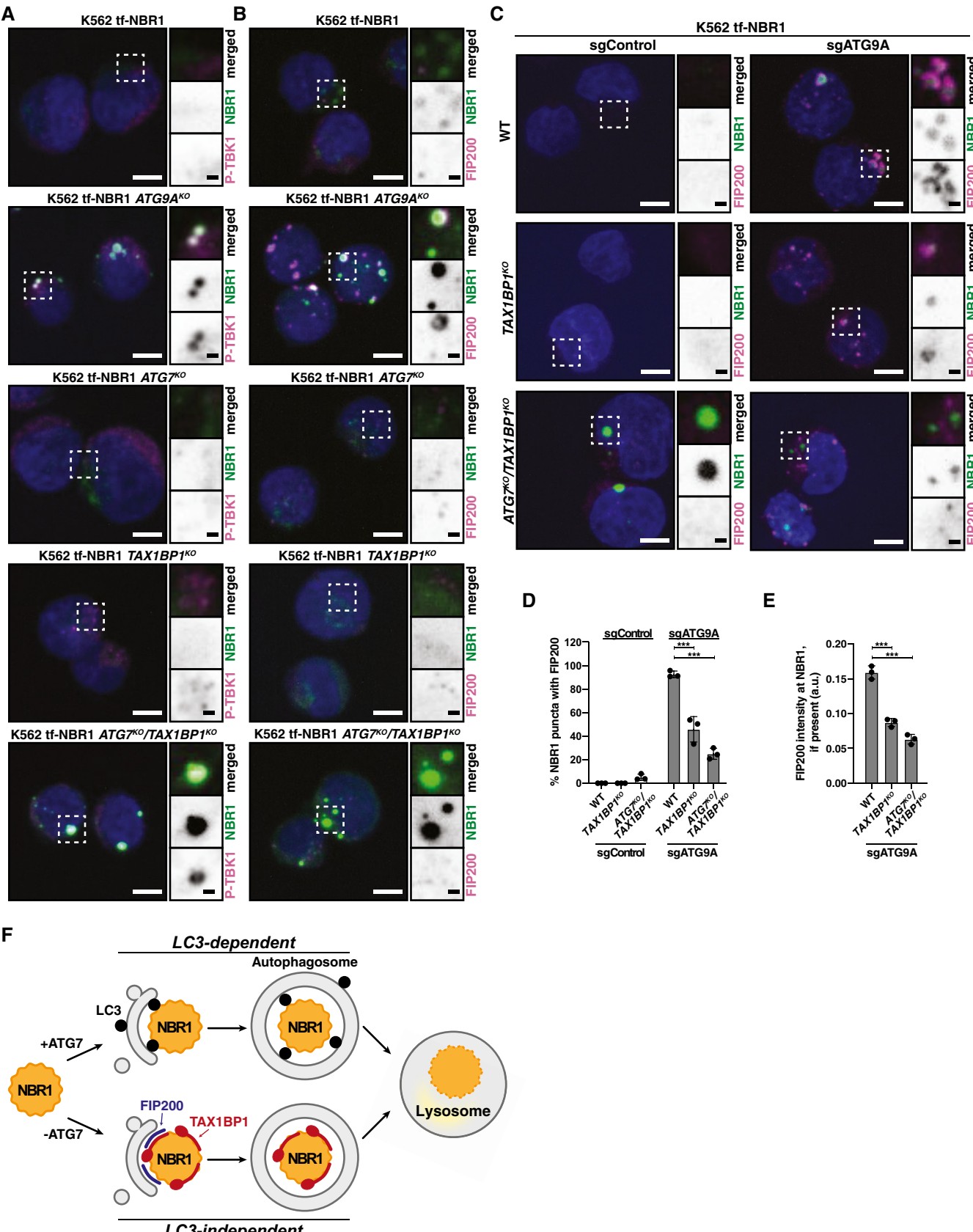

**Figure 7.**

◄

Figure 7.  TAX1BP1 recruits FIP200 to induce local autophagosome formation.

A   Representative confocal micrographs (as maximum intensity projections) from wild-type and deletion K562 cells expressing tf-NBR1. Selected regions (white box) of micrographs are shown as single and merged channels from fluorescence and immunofluorescence microscopy against indicated proteins. P-TBK1, magenta; NBR1, green; Hoechst, blue. Scale bars: large panels, 5 µm; small panels, 1 µm.

B   Representative confocal micrographs (as maximum intensity projections) from wild-type and deletion K562 cells expressing tf-NBR1. Selected regions (white box) of micrographs are shown as single and merged channels from fluorescence and immunofluorescence microscopy against indicated proteins. FIP200, magenta; NBR1, green; Hoechst, blue. Scale bars: large panels, 5 µm; small panels, 1 µm.

C   Representative confocal micrographs (as maximum intensity projections) from wild-type, *TAX1BP1^KO*, and *ATG7^KO*/*TAX1BP1^KO* K562 cells expressing tf-NBR1 and transduced with indicated sgRNA. Selected regions (white box) of micrographs are shown as single and merged channels from fluorescence and immunofluorescence microscopy against indicated proteins. FIP200, magenta; NBR1, green; Hoechst, blue. Scale bars: large panels, 5 µm; small panels, 1 µm.

D   Quantitation of colocalization between NBR1 with FIP200 in wild-type, *TAX1BP1^KO*, and *ATG7^KO*/*TAX1BP1^KO* cells imaged in (C) (see Materials and Methods for details of quantitation). Bar graphs represent mean ± SD for three independently generated deletion cell lines (dots). sgATG9A samples were compared using a one-way ANOVA ($P < 0.0001$) with Tukey's HSD post-test. ***$P < 0.001$. $n > 200$ NBR1 puncta for each biological replicate.

E   Quantitation of FIP200 intensity at NBR1 puncta in wild-type, *TAX1BP1^KO*, and *ATG7^KO*/*TAX1BP1^KO* cells imaged in (C). Bar graphs represent mean ± SD for three independently generated deletion cell lines (dots). $n > 80$ FIP200-positive puncta for each biological replicate. Samples were compared using a one-way ANOVA ($P < 0.0001$) with Tukey's HSD post-test. ***$P < 0.001$. a.u., arbitrary intensity units.

F   TAX1BP1 drives local autophagosome formation in the context of NBR1 puncta. This activity is independent of LC3 or the ubiquitin-binding domains of TAX1BP1. Rather, a newly defined NBR-interacting domain (or N domain, not shown) drives the association of TAX1BP1 (red) with NBR1 puncta. Subsequent recruitment and clustering of FIP200 (blue) by TAX1BP1 induces local autophagosome formation and enforces cargo specificity, thereby replacing the requirement for lipidated LC3.

Data information: See also Fig EV5.
Source data are available online for this figure.

ATG7-independent autophagy and a facilitator of it. The ATG7-independent function of TAX1BP1 uniquely depends on several features including its SKICH domain and a newly identified N domain, which associates with NBR1. In particular, we find that a single point mutation within the SKICH domain (TAX1BP1^A114Q), which disrupts binding to both TBK1 and FIP200, is sufficient to impede *in situ* autophagosome formation during ATG7-independent autophagy. In contrast, TAX1BP1^A114Q has minimal effect on canonical autophagy. Furthermore, while much is written about the ubiquitin-binding properties of SQSTM1-family receptors, ATG7-independent autophagy did not require TAX1BP1's UBZ domains, possibly because the N domain (defined here) provides an alternative means for receptor clustering. By comparison, the UBZ domains strongly influence the ability of TAX1BP1 to be incorporated by into canonical autophagosomes. We anticipate that these tools, which genetically separate the canonical and non-canonical roles of TAX1BP1 (and by extension, likely other receptors), will facilitate a more rapid and comprehensive understanding of autophagosome biogenesis and receptor function.

Cargo-induced autophagosome formation around autophagy targets is increasingly understood to be driven by autophagy receptor proteins through local clustering of FIP200 (ATG11 in yeast) and TBK1 activation (Itakura *et al*, 2012; Heo *et al*, 2015; Kamber *et al*, 2015; Lazarou *et al*, 2015; Richter *et al*, 2016; Torggler *et al*, 2016; Smith *et al*, 2018; Ravenhill *et al*, 2019; Turco *et al*, 2019; Vargas *et al*, 2019; Turco *et al*, 2020). In mammalian systems, this mechanism has been reported for Pink/Parkin-mediated mitophagy and xenophagy, dependent on NDP52, and for ubiquitin aggregates, dependent on SQSTM1 (Lazarou *et al*, 2015; Ravenhill *et al*, 2019; Turco *et al*, 2019; Vargas *et al*, 2019). However, the prevalence of cargo-induced autophagosome formation mechanisms in mammalian cells remains unknown. Here, we report that TAX1BP1 is able to drive local autophagosome formation in the context of NBR1 puncta. Moreover, we identify the LC3-independent, UBZ-independent, adapter function of TAX1BP1 as sufficient for this *in situ* autophagosome formation. Thus, it seems increasingly likely that local autophagosome formation through FIP200 clustering is a

general principle governing autophagy receptor function. This activity is sufficient to generate functional autophagosomes and enforces cargo specificity in the absence of LC3. As such, these data provide a partial mechanistic rationale for previous reports of ATG7-independent autophagy.

We note that, unlike in previous reports, NDP52 remains diffuse in our experimental system, likely due to a lack of targets or other nucleating signals. On the other hand, NBR1 forms puncta but is insufficient to recruit FIP200 or drive *in situ* autophagosome biogenesis. Rather, either TAX1BP1 or LC3-lipidation are also required for NBR1 flux. These data point toward an increasingly complex network of interactions—involving feedback loops, signal amplification, and receptor plasticity—that ensure a robust autophagy response (Lazarou *et al*, 2015; Richter *et al*, 2016; Padman *et al*, 2019). As demonstrated here and by others, the use of orthologous autophagy reporters, synthetic genetic interactions, and separation-of-function mutants can be used to deconvolve concomitant mechanisms within autophagy and will undoubtedly yield further insights in the future. Specifically, important future goals will be to determine (i) the relative contribution of LC3-dependent and LC3-independent mechanisms in unmodified cells and (ii) additional physiological instances in which LC3 lipidation is limiting for autophagy (e.g., *Legionella* infection).

Within other modifiers of ATG7-independent autophagy that we identified, KAT8 (alternatively hMOF or MYST1) is an H4K16 lysine acetyltransferase previously implicated in the regulation of genes required for autophagy and lysosomal biogenesis (Füllgrabe *et al*, 2013; Sheikh *et al*, 2016). Similarly, TRAPPC11, as a member of the TRAPP III complex, has been previously implicated in autophagy regulation (Stanga *et al*, 2019). GDI2 is a Rab GDP dissociation inhibitor (Rab-GDI) (Yang *et al*, 1994). In this capacity, GDI2 is thought to regulate Rab function through modulating nucleotide occupancy. We validated the role of GDI2 in ATG7-independent autophagy; however, its specific function remains to be elucidated. It is tempting to speculate that GDI2 may compensate for other functions ascribed to lipidated LC3, such as membrane expansion or autophagosomal trafficking. Lastly, TBK1, a kinase associated with TAX1BP1, was also identified in our screen. Multiple roles have

been proposed for TBK1 in autophagy, but TBK1 activation is frequently observed in instances of local autophagosome formation (Heo *et al*, 2015; Matsumoto *et al*, 2015; Richter *et al*, 2016; Ravenhill *et al*, 2019; Vargas *et al*, 2019). Here, we report that the interaction between TBK1 and TAX1BP1 is required for ATG7-independent flux of NBR1, although the recruitment of TBK1 to NBR1 puncta does not strictly require TAX1BP1. Future work will be required to fully dissect the contributions of TBK1 to local autophagosome formation.

In summary, our systematic analysis casts a new light on the roles of ATG factors within the autophagy cannon. Specifically, the data herein provide a partial mechanistic basis for reports of selective autophagy in cells lacking the lipidation machinery (e.g., *ATG7*[KO] or *ATG5*[KO] cells). Furthermore, we identified a ubiquitin-independent mode of TAX1BP1 recruitment through its N domain, highlighting that TAX1BP1 recruitment and clustering, rather than ubiquitin binding *per se*, is critical for receptor function. Collectively, our data reinforce the duality of mammalian autophagy receptors, in this case TAX1BP1, to both tether cargo to autophagic membranes (via LC3) and/or, independently, recruit upstream autophagy factors to drive local autophagosome formation. Practical applications of this finding relate to the limited experimental utility of using *ATG7*[KO] cells, or similar genetic variants, as lone autophagy-deficient controls. In addition, clinical development of lipidation inhibitors (e.g., ATG4 or ATG7 inhibitors) will need to account for the differential degradation of etiological substrates with and without lipidation.

# Materials and Methods

## Antibodies

For immunoblotting (IB), all primary antibodies were diluted 1:1,000 unless otherwise noted. All secondary antibodies were diluted 1:10,000. For immunofluorescence (IF), primary antibodies were diluted 1:100 unless otherwise noted; secondary antibodies were diluted 1:1,000. The following primary antibodies were used: rabbit anti-NBR1 (A305-272A, Bethyl Labs), rabbit anti-TAX1BP1 ({1:200 —IF} 5105, CST), mouse anti-SQSTM1 (ab56416, Abcam), rabbit anti-NDP52 (9036, CST), rabbit anti- LC3A/B (12741S, CST), rat anti-tubulin ({1:500—IB} sc-53030, Santa Cruz), rabbit anti-ATG7 (8558, CST), rabbit anti-TROVE2 (ab194004, Abcam), rabbit anti-ATG5 (12994S, CST), rabbit anti-ATG9A (13509S, CST), rabbit anti-FIP200 (12436S, CST), rabbit anti-V5 (13202, CST), mouse anti-myc (M4439, Sigma), rabbit anti-HA (3724T, CST), rabbit anti-TBK1 (3013S, CST), rabbit anti-P-TBK1 (5483S, CST), mouse anti-GFP (11814460001, Sigma), mouse anti-TagBFP (EVN-AB234-C200, Axxora); secondary antibodies (IB): goat anti-mouse IgG(H + L) IRDye 680LT (926-68020, LI-COR), goat anti-rabbit IgG(H + L) IRDye 800CW (926-32211, LI-COR); secondary antibodies (IF): goat anti-rabbit IgG(H + L) Alexa Fluor Plus 647 (A32733, Invitrogen), goat anti-mouse IgG(H + L) Alexa Fluor Plus 647 (A32728, Invitrogen).

## Chemicals and reagents

The following chemicals and reagents were used in this study: 2-mercaptoethanol (BME) (M6250-100ML, Sigma), agar (A10752, Alpha Aesar), agarose (16500500, Thermo Fisher), ampicillin (A9518-25G,

Sigma), Bafilomycin A1 (11038, Caymen chemical), blasticidin (ant-bl-1, Invivogen), EDTA (EDS-500G, Sigma), glycerol (G2025-1L, Sigma), HEPES (H3375-1KG, Sigma), kanamycin (BP906-5, FisherSci), normocin (ant-nr-1, Invivogen), Phusion High-Fidelity DNA polymerase (M0530L, NEB), polybrene (H9268-5G, Sigma), potassium chloride (P217-500, FisherSci), puromycin (ant-pr-1, Invivogen), sodium chloride (6438, FisherSci), sodium dodecyl sulfate (SDS) (74255-250G, Sigma), sucrose (BP220-1, FisherSci), Taq DNA ligase (M0208L, NEB), Tris base (T1378-5KG, Sigma), TritonX-100 (T9284-500ML, Sigma), tryptone (DF0123-17-3, FisherSci), Tween-20 (BP337-500, FisherSci), T5 exonuclease (M0363S, NEB), yeast extract (BP1422-2, FisherSci), and zeocin (ant-zn-1, Invivogen).

## Vectors

The Brunello knockout pooled library was a gift from David Root and John Doench (Addgene #73178). psPAX2 was a gift from Didier Trono (Addgene plasmid # 12260). pCMV-VSV-G was a gift from Bob Weinberg (Addgene plasmid #8454). lentiCRISPRv2 puro was a gift from Brett Stringer (Addgene plasmid #98290). lentiGuide-puro was a gift from Feng Zhang (Addgene plasmid #52963). pFUGW-EFSp-Cas9-P2A-Zeo (pAWp30) was a gift from Timothy Lu (Addgene plasmid #73857). pLenti CMV GFP Puro (658-5) was a gift from Eric Campeau & Paul Kaufman (Addgene plasmid #17448). pHAGE-myc-AP2-TAX1BP1 was a gift from Christian Behrends). pCMV-hATG7WT and pCMV-hATG7CS were gifts from Eiki Kominami & Isei Tanida (Addgene plasmids #87867 and #87868). Other vectors generated in the course of this study are available upon request.

## Isothermal assembly

PCR inserts were amplified using Phusion High-Fidelity DNA polymerase (M0530L, NEB). Amplification primers were designed to append a 30 base pair overlap with the linear ends of restriction-digested recipient vectors. Linearized vector backbones were dephosphorylated by calf intestinal phosphatase (M0290S, NEB). All inserts and vectors were purified from a 0.9% agarose gel prior to isothermal assembly (D4002, Zymo Research). 50 ng of linearized vector DNA was combined with isomolar amounts of purified insert (s). 2.5 µl DNA mix was incubated with 7.5 µl isothermal assembly master mix at 50°C for 20 min. Product of the isothermal assembly reaction was transformed into NEB Stable cells (C3040H, NEB). Transformed cells were plated on plates of LB media (10 g/l tryptone, 5 g/l yeast extract, 5 g/l NaCl) containing 1.5% agar. 100 µg/ml ampicillin or 50 µg/ml kanamycin were included in bacterial cultures, where appropriate. All cultures and plates were grown overnight at 34°C. Overnight cultures were pelleted at 3,000 *g* for 10 min and plasmid DNA was purified using a Qiagen miniprep kit (27106, Qiagen). Sequences were verified by Sanger sequencing (Eton Bioscience Inc).

## sgRNA oligonucleotide ligation protocol

Oligonucleotides were ordered from Thermo Fisher. For sgRNA cloning, oligos were ordered in the following format: Forward: 5′-CACCGNNNNNNNNNNNNNNNNNNNN-3′; Reverse: 5′-AAACN NNNNNNNNNNNNNNNNNNNC-3′. 50pmol of each oligo were

mixed in a 25 µl reaction and phosphorylated with T4 polynucleotide kinase (M0201S, NEB). Reactions were performed for 30 min at 37°C in 1X T4 DNA ligase buffer (B0202S, NEB). Phosphorylated oligos were annealed by heating for 5 min at 95°C and slow cooling (0.1°C/s). 2 µl of diluted (1:100) oligo mix was ligated into 20 ng BsmBI-digested vector (pLentiGuide-puro or pLenti-CRIPSR v2) using T4 DNA ligase (M0202S, NEB). Ligation was performed at room temperature for 15 min.

## Tissue culture

K562 cells expressing tf-NBR1, tf-TAX1BP1, tf-NDP52, tf-SQSTM1, and tf-LC3 were generated previously (Shoemaker *et al*, 2019). All cells were grown in a standard water-jacketed incubator with 5% $CO_2$. K562 cells were grown in IMDM media (10-016-CV, Corning) with 10% FBS (26140079, Thermo Fisher) and 1× penicillin/strep (15140122, Thermo Fisher). Cells were maintained below 1 million cells per milliliter. HEK293T cells were grown in DMEM media (10-013-CV, Corning) with 10% FBS and 1× pen/strep. Normocin (1:500) was used as a common additive. All cells were passaged < 25 times. For passaging, cells were trypsinized with Trypsin-EDTA (25300-054, Gibco). Puromycin (2 µg/ml), blasticidin (5 µg/ml), and zeocin (50 µg/ml) were added when necessary for selection. HBSS was used to wash cells (14025092, Thermo Fisher).

## Cell line authentication

Genomic DNA was isolated from HEK293T and K562 cells using the GenElute Mammalian Genomic DNA Miniprep Kits (Sigma-Aldrich). STR profiling and allele identification were performed by the Molecular Diagnostics Laboratory of Dana-Farber Cancer Institute. Briefly, isolated genomic DNA was analyzed with the GenePrint 10 short tandem repeat (STR) profiling kit (Promega) and Amelogenin for gender identification. GeneMapper v4 Fragment Analysis software (Applied Biosystems) and GenePrint10 allele panel (Promega) custom bin files were used to identify the alleles at eight STR loci (TH01, TPOX, vWA, CSF1PO, D16S539, D7S820, D13S317, and D5S818). The ATCC STR Profile Database was used to verify that the identified alleles matched those of the expected cell type.

## Transient transfection and nucleofection

HEK293T cells were grown overnight in Opti-MEM Reduced Serum media with 5% FBS (51985-034, Thermo Fisher). When 90% confluent, cells were transfected using Lipofectamine 3000 reagent (L3000008, Life Technologies) according to the manufacturer's recommendations. K562 cells were nucleofected using a Nucleofector™ 2b Device (Lonza) using nucleofector kit T (VACA-1002, Lonza) and protocol T-016.

## Generation of gene knockout cell lines using CRISPR-Cas9 gene editing

Sequences for sgRNAs targeting genes of interest were extracted from the Brunello library and cloned into the indicated vectors as outlined above under "sgRNA oligonucleotide ligation protocol". HEK293T and K562 cells were transfected or nucleofected, respectively, with the resulting vectors. Limiting dilution or cell sorting was used to isolate individual clones. Knockout of expanded clones was confirmed by immunoblot.

## Lentiviral generation

Lentivirus was generated in HEK293T cells using Lipofectamine 3000 (L3000008, Life Technologies). Cells were grown overnight in Opti-MEM media (5% FBS, no antibiotics) (51985-034, Thermo Fisher) to 90% confluency. Cells were transfected with pVSV-G, pSPAX2 and expression constructs at a 1:4:3 ratio. Transfection proceeded for 6–8 h before media was refreshed. Virus was collected and pooled at 24 and 48 h post-transfection. Virus was pelleted at 1,000 $g$ 2× 10 min, aliquoted and frozen in single-use aliquots.

## Viral transduction

Cells were incubated in appropriate media containing 8 µg/ml polybrene and lacking penicillin/streptomycin. Cells were transduced overnight. In the morning, virus-containing media were exchanged for fresh media lacking polybrene. Cells were allowed to recover for 24 h prior to antibiotic selection.

## Protease protection assay

K562 cells were exchanged into fresh IMDM media containing 75 nM Bafilomycin A1 and incubated for 18 h. After incubation, cells were pelleted, washed 1× with cold HBSS (14025092, Thermo Fisher), and resuspended in pre-chilled lysis buffer (20 mM Hepes KOH pH7.4, 0.22 M mannitol, 0.07 M sucrose). Cells were lysed by extrusion through a 26 gauge needle 20 times. Samples were pelleted at 150 g for 5 min at 4°C to pellet debris. When indicated, samples were incubated with 1× trypsin (T1426-100MG, Sigma; 100× stock: 2 mg/ml) and/or 0.5% Triton X-100 for 90 min at 37°C. Reactions were quenched in 1× hot Laemmli sample buffer and held at 65°C for 10 min.

## Immunoprecipitation

Cells were collected and resuspended in lysis/IP buffer (50 mM HEPES pH 7.4, 150 mM NaCl, 2 mM EDTA, 1% Triton X-100, 2× cOmplete protease inhibitor tablet [5056489001, Sigma]) or IP Buffer + 5% glycerol. Cells were incubated on ice for 30 min and pelleted twice at 20,000 $g$ for 10 min at 4°C. Supernatant was normalized by total protein using a BCA assay prior to IP. Normalized extract was applied to GFP-Trap dynabeads (gtd-10, Chromotek) protein G dynabeads (10003D, Thermo Fisher) pre-bound to the indicated antibodies. Incubation was allowed to proceed for 1 h at 4°C. Beads were washed 4× with two tube changes. Protein was eluded by boiling at 70°C in 1× Laemmli Loading Buffer (3× stock: 189 mM Tris pH 6.8, 30% glycerol, 6% SDS, 10% beta-mercaptoethanol, bromophenol blue).

## Gel electrophoresis and immunoblotting

Cells were lysed for 15 min on ice in lysis buffer (50 mM HEPES pH 7.4, 150 mM NaCl, 2 mM EDTA, 1% Triton X-100, 2× cOmplete protease inhibitor tablet (5056489001, Sigma)). Lysates were cleared 2× at 500 $g$ for 5 min. Post-spin supernatants were

used as input. Protein levels in supernatants were normalized using a BCA protein assay (#23227, Pierce). Normalized samples were boiled in 1× (final concentration) Laemmli Loading Buffer. Gel electrophoresis was performed at 195V for 65 min in Novex 4–20% Tris-Glycine gels (WXP42020BOX, Thermo Fisher). For Western blotting, samples were transferred for 60 min to 0.2 µm PVDF membranes (#ISEQ00010, Sigma) using a Semi-dry transfer cell (Bio-Rad). Membranes were blocked for 20 min in Intercept™ (TBS) Blocking Buffer (927-60003, LI-COR). Primary antibodies were incubated overnight at 4°C. Blots were then rinsed 3× 5 min in TBS-T (10 mM Tris pH 7.9, 150 mM NaCl, 0.05% Tween-20, 0.25 mM EDTA). Secondary antibodies (1:10,000 dilution in Intercept™ [TBS] Blocking Buffer with 0.2% Tween-20 and 0.01% SDS) were incubated for 1 h at room temperature. Blots were rinsed 4× 10 min in TBS-T and imaged using fluorescence (LI-COR Odyssey CLx Imager).

## Immunofluorescence/immunocytochemistry

Cells were fixed for 15 min in 4% paraformaldehyde (PFA) made from fresh 16% PFA (#15710, Electron Microscopy Sciences) diluted with 1× Dulbecco's phosphate buffered saline with calcium chloride and magnesium chloride (14080-055, Thermo Fisher). Cells were pelleted for 5 min at 150 $g$, PFA was removed, and cells were washed twice with 1× DPBS. Cells were blocked at RT for 1 h in Intercept™ (TBS) Blocking Buffer (927-60003, LI-COR) plus 0.3% Triton X-100, then washed once in 1× DPBS. Primary antibody was diluted in Intercept™ (TBS) Blocking Buffer, and cells were incubated in 100 µl of primary antibody solution overnight at 4°C. After incubation, cells were washed 3× 5 min in 1× DPBS. Secondary antibody was diluted to 1:1,000 in Intercept™ (TBS) Blocking Buffer, and cells were incubated in 100 µl of secondary antibody solution for 45 min at RT. After incubation, cells were washed 3× 10 min in 1× DPBS, stained with a 1:10,000 dilution of Hoechst 33342 (H3570, Thermo Fisher) for 5 min, and washed once more in 1× DPBS before mounting on slides (294875X25, Corning) using Prolong Diamond (P36965, Thermo Fisher) and #1.5 Gold Seal coverglass 22 × 22 MM (63786-01, Electron Microscopy Science).

## Confocal microscopy

Fluorescent images were obtained using an Andor W1 Spinning Disk Confocal on a Nikon Eclipse Ti inverted microscope and a 40X PlanAPO objective lens (Nikon).

## Image analysis

Images intensities were modified linearly and evenly across samples. Maximum intensity projections were generated in ImageJ. Channels were individually imported into CellProfiler 3.1.9. Primary objects were identified based on nuclear staining. Punctate structures were identified using "IdentifyPrimaryObjects". Object intensity, size, and colocalization were quantified automatically using MeasureObjectIntensity, MeasureObjectShapeSize, and RelateObjects, respectively. All quantitation was performed in a blinded, automated fashion. All reported data are the median of three biological replicates (i.e., independently generated cell lines).

## Library propagation

Brunello library (two vector system) was purchased from Addgene (#73178). 50ng of library was electroporated into 25 µl Endura electrocompetent cells (60242-2, Lucigen). Cells from eight electroporations were pooled and rescued in 8 ml of rescue media for 1 h at 37°C. 8 ml of SOC (2% tryptone, 0.5% yeast extract, 10 mM NaCl, 2.5 mM KCl, 10 mM MgCl$_2$, 10 mM MgSO$_4$, and 20 mM glucose) was added to cells and 200 µl of the final solution was spread onto 10 cm LB plates containing 50 µg/ml carbenicillin (80 plates total). Cells were manually scraped off plates, and a GenElute Megaprep kit (NA0600-1KT, Sigma) was used to purify plasmid DNA.

## Library lentiviral generation

Lentivirus was generated by lipofection (L3000008, Life Technologies) of HEK293T cells with 5 µg psPAX2 (Addgene Plasmid #12260), 1.33 µg pCMV-VSV-G (Addgene plasmid Plasmid #8454), and 4 µg library vector per 10 cm plate. Low-passage HEK293T cells were grown in Opti-MEM + 5% FBS. Cells were grown to 95% confluency and transfected for 6 h. Media were then replaced with fresh Opti-MEM + 5% FBS. Twenty-four hours post-transfection, supernatant was collected and replaced. Forty-eight hours post-transfection, supernatant was again collected, pooled with the 24 h supernatant and clarified 2× 1,000 $g$ for 10 min.

## Transduction and cell growth

For CRISPR screening experiments, K562 cells were passaged to maintain cell density between 500,000 and two million cells/ml. Cells were propagated in IMDM + 10% FBS + pen/strep + appropriate antibiotics (Blasticidin 5 µg/ml, zeocin 50 µg/ml) until 200 million cells were obtained (approximately 8–10 days). 200 million cells were pelleted and resuspended in IMDM + 10% FBS + 8 µg/ml polybrene. An MOI of 0.4 was used to minimize multiple infection events per cell. Date of infection was day 0. Cells were infected overnight, pelleted and exchanged into fresh media. After 24 h, cells were split and 2 µg/ml puromycin was added. Cells were continually passaged in puromycin. At day 9, cells were removed from puromycin and at day 10–12 cells were sorted for red:green fluorescence. 50 M unsorted cells were pelleted and processed as input. The top and bottom 30% of cells (based on Red:Green ratio) were taken. One hundred million cells were sorted for each experimental condition. Cell sorting was performed using a Sony SH800 cell sorter. Cells were pelleted and stored at −80°C until processing.

## RNA extraction, cDNA synthesis, and qRT–PCR

Total RNA was extracted using TRIzol reagent according to the manufacturer's specifications (15596026, Thermo Fisher). cDNA was synthesized from 900 ng of RNA using the Invitrogen™ SuperScript™ IV First-Strand Synthesis System (18091050, Thermo Fisher) and an oligo(dT) primer. The resulting cDNAs were treated with RNAse H and diluted 10-fold. Diluted cDNA was combined with PowerUP™, SYBR Green PCR master mix (A25742, Applied Biosystems), and 0.4 µM of both forward and reverse primers. PCR was performed using a StepOne Plus qRT–PCR machine (Applied Biosystems). Quantitation of each mRNA (GAPDH, RPL0 and NBR1) was

performed in triplicate and normalized to GAPDH expression using the comparative Ct method.

## CRISPR screen processing

Genomic DNA was purified from collected cells using the NucleoSpin Blood XL kit (740950.1, Machery Nagel) according to the manufacturer's instructions. sgRNA sequences were amplified from total genomic DNA using a common pool of eight staggered-length forward primers. Unique 6-mer barcodes within each reverse primer allowed multiplexing of samples. Each 50 µL PCR reaction contained 0.4 µM of each forward and reverse primer mix (Integrated DNA Technologies), 1× Phusion HF Reaction Buffer (NEB), 0.2 mM dNTPs (NEB), 40 U/ml Phusion HF DNA Polymerase (NEB), up to 5 µg of genomic DNA, and 3% v/v DMSO. The following PCR cycling conditions were used: 1× 98°C for 30 s; 25× (98°C for 30 s, 56°C for 30 s, 63°C for 30 s); 1× 63°C for 10 min. The resulting products were pooled to obtain the sgRNA libraries. The pooled PCR products were size selected between 0.60× and 0.85× magnetic bead slurry as outlined by DeAngelis *et al* (1995). Library purity and size distribution was measured on a Fragment Analyzer instrument (Agilent) and quantified fluorometrically by Qubit. Libraries were pooled in equimolar ratios and loaded at 2.5 pM onto a NextSeq500 High Output 75cycle run. 2% PhiX spike in was included as an internal control for sequencing run performance. Data were demultiplexed into fastq files using Illumina bcl2fastq2 v2.20.0.422.

## NGS data analysis

The 5′ end of Illumina sequencing reads was trimmed to 5′-CACCG-3′ using Cutadapt (Martin, 2011). The count function of MAGeCK (version 0.5.9) was used to extract read counts for each sgRNA (Li *et al*, 2014). Raw read counts can be found in Table EV1. The mle function was used to compare read counts from cells displaying increased and decreased Red:Green ratios (Li *et al*, 2015). The output included both beta scores and false discovery rates. All beta scores can be found in Table EV2. Beta scores for each sgRNA for each tf-Reporter were averaged across 2–4 experiments. Across all experiments, average read counts were 200–400 per sgRNA. To generate heat maps for each reporter (e.g., Fig 1C), the beta scores for each gene were normalized by the beta score for ATG9A.

## Flow cytometry

Cells were trypsinized (if adherent), pelleted, washed in cold HBSS, and filtered through 41-µm nylon mesh (B0015GZDEQ, Amazon) prior to analysis. Data were collected on a MACSQuant VYB flow cytometer. Data were analyzed using FlowJo v 10 (FlowJo, LLC) and R. The biomodality of populations was determined in an unbiased manner using the BifurGate tool in FlowJo. At least 5,000 cells were collected for all samples.

## Statistical analysis

Statistical analysis was performed using Prism 8 (GraphPad). All statistical tests are indicated in the relevant figure legends. All tests were two-tailed with $P < 0.05$ as the threshold for statistical significance. Prior to performing the indicated test(s), Shapiro–Wilk was used to test normality. Brown–Forsythe test was used to test unequal variance for ANOVA. *F*-test for equality of variances was used for *t*-tests. Number of replicates (*n*) used for each test are indicated in the relevant figure legend.

## Correlative light and electron microscopy (CLEM)

For CLEM, K562 $ATG7^{KO}$, or $ATG7^{KO}/TAX1BP1^{KO}$ cells, stably expressing tf-NBR1 were seeded on poly-lysine coated, photo-etched coverslips (Electron Microscopy Sciences, Hatfield, USA) and left to adhere for 3–4 h. Cells were fixed in 2% glutaraldehyde/0.1 M PHEM (60 mM PIPES, 25 mM HEPES, 2 mM MgCl₂, 10 mM EGTA), pH 6.9, for 1 h. The coverslips were washed with 0.1M PHEM buffer and mounted with Mowiol containing 1 µg/ml Hoechst 33342. The cells were examined with a Zeiss LSM710 confocal microscope (Carl Zeiss MicroImaging GmbH, Jena, Germany) utilizing a Laser diode 405-30 CW (405 nm), an Ar-Laser Multiline (458/488/514 nm) and a DPSS-561 10 (561 nm). Cells of interest were identified by fluorescence microscopy, and a Z-stack covering the whole cell volume was acquired. The relative positioning of the cells on the photo-etched coverslips was determined by taking a low magnification DIC image. The coverslips were removed from the object glass, washed with 0.1 M PHEM buffer, and fixed in 2% glutaraldehyde/0.1 M PHEM over night. Cells were postfixed in osmium tetroxide and potassium ferry cyanide, stained with tannic acid, and uranyl acetate, and thereafter dehydrated stepwise to 100% ethanol followed by flat-embedding in Epon. Serial sections (200 nm) were cut on a Ultracut UCT ultramicrotome (Leica, Germany) and collected on formvar coated slot-grids. Sections were observed at 200 kV in a Thermo ScientificTM TalosTM F200C microscope and recorded with a Ceta 16M camera. Consecutive sections were used to align electron micrographs with fluorescent images in *X*, *Y*, and *Z*. For tomograms, image series were taken between −60° and 60° tilt angles with 2° increment. Single-tilt axes series were recorded with a Ceta 16 M camera. Tomograms were computed using weighted back projection using the IMOD package. Display, segmentation and animation of tomograms were also performed using IMOD software version 4.9 (Kremer *et al*, 1996).

## Mass spectroscopy

Proteins were enriched using the SP3 method (Hughes *et al*, 2019) and digested overnight with trypsin. Peptides were desalted and analyzed by label-free LC-MS/MS analysis on a Q-Exactive Plus quadrupole Orbitrap mass spectrometer (Thermo Scientific) equipped with an Easy-nLC 1000 (Thermo Scientific) and nanospray source (Thermo Scientific) as previously described (Rusin *et al*, 2015). Raw data were searched using COMET in high resolution mode (Eng *et al*, 2013) against a target-decoy (reversed) (Elias & Gygi, 2007) version of the human proteome sequence database (UniProt; downloaded 2/2013, 40,482 entries of forward and reverse protein sequences) with a precursor mass tolerance of ± 1 Da and a fragment ion mass tolerance of 0.02 Da, and requiring fully tryptic peptides (K, R; not preceding P) with up to three mis-cleavages. Static modifications included carbamidomethylcysteine and variable modifications included: oxidized methionine. Searches were filtered using orthogonal measures including mass measurement accuracy

($\pm$ 3 ppm), Xcorr for charges from +2 through +4, and dCn targeting a < 1% FDR at the peptide level. Quantification of LC-MS/MS spectra was performed using MassChroQ (Valot *et al*, 2011) and the iBAQ method (Schwanhäusser *et al*, 2011). Missing values were imputed from a normal distribution in Perseus to enable statistical analysis (Tyanova *et al*, 2016). Statistical analysis was carried out in Perseus by two-tailed Student's *t*-test.

## Data availability

CRISPR screening data are presented in Tables EV1 and EV2. Mass spectroscopy data are presented in Table EV3. Raw MS data for the experiments performed in this study are available at MassIVE (PXD021363; https://massive.ucsd.edu/ProteoSAFe/dataset.jsp?task= 7fd74db91b1a4d2787be4d0bd7231fc3). Source data and quantifications given in the main text have associated raw data.

**Expanded View** for this article is available online.

## Acknowledgements

We thank Ann Lavanway and members of the BioMT facility, the Genomics and Molecular Biology Shared Resource, and the DartLab facility for technical support with imaging, flow cytometry, and Illumina sequencing; Vladimir Denic, Bill Wickner, and Michael Ragusa for critical reading of the manuscript; Marianne Smestad and Ulrikke Dahl Brinch for technical support; and members of the department for scientific advice. This work was supported by the National Institutes of Health R00GM117218 (to C.J.S.), R35GM119455 (to A.N.K), P20GM113132 (to Dartmouth BioMT), P30CA023108 (to Norris Cotton Cancer Center), and the Geisel School of Medicine.

## Author contributions

Investigation, Validation, Visualization, Writing—review & editing: AEO, JMD, BJN, CJS, SWS, IN, ANK. Methodology, Formal Analysis: AEO, CJS, IN, ANK. Writing—original draft, Conceptualization, Funding acquisition, Supervision: CJS.

## Conflict of interest

The authors declare that they have no conflict of interest.

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
