## [Review Process File · The EMBO Journal]

Receptor-mediated clustering of FIP200 bypasses the role of LC3 lipidation in autophagy

Amelia Ohnstad, Jose Delgado, Brian North, Isha Nasa , Arminja Kettenbach, Sebastian Schultz, and Christopher Shoemaker

DOI: [10.15252/embj.2020104948](https://doi.org/10.15252/embj.2020104948)

Corresponding author(s): Christopher Shoemaker (Christopher.J.Shoemaker@Dartmouth.edu)

Review Timeline:

Submission Date:	9th Mar 20
Editorial Decision:	28th Apr 20
Revision Received:	11th Aug 20
Editorial Decision:	7th Sep 20
Revision Received:	10th Sep 20
Accepted:	17th Sep 20

Editor: Elisabetta Argenzio

Transaction Report:

Thank you for submitting your manuscript entitled "Receptor-mediated clustering of RB1CC1 bypasses the role of LC3 lipidation in autophagy" to The EMBO Journal. Please accept my apologies for the lengthy review process. Your study has been sent to three reviewers for evaluation, whose reports are enclosed below.

As you can see, the referees consider the work potentially interesting. However, they also raise several major criticisms that need to be addressed before they can support publication in The EMBO Journal. Importantly, referee #1 stresses that the physiological relevance of ATG7-independent autophagy and the proposed ubiquitin-independent recruitment of TAX1BP1 to cargoes would need to be further investigated.

Given the overall interest of your study, I would like to invite you to submit a revised version that addresses the above-mentioned and all the other points raised by the referees. I would like to point it out that solving all the referees' issues in a conclusive manner will be essential for publication in The EMBO Journal, as well as a strong support from the referees. I should also add that it is our policy to allow only a single round of revision. Therefore, acceptance of your manuscript will depend on the completeness of your responses in this revised version.

We generally allow three months as a standard revision time. As we are aware that many laboratories cannot function at full capacity owing to the COVID-19 pandemic, we may relax this deadline. Also, we have decided to apply our 'scooping protection policy' to the time span required for you to fully revise your manuscript and address the experimental issues highlighted herein. Nevertheless, please inform us as soon as a paper with related content published elsewhere.

Before submitting your revision, primary datasets (and computer code, where appropriate) produced in this study need to be deposited in an appropriate public database (see <http://msb.embopress.org/authorguide#dataavailability>). Please remember to provide a reviewer password if the datasets are not yet public. Include a "Data availability" section even if there are no primary datasets produced in the study.

I realize that addressing all the referees' criticisms could be time-consuming and technically challenging. I would therefore understand if you were to choose not to undergo an extensive revision here and rather submit your study elsewhere, in which case please inform us at your earliest convenience.

I thank you for the opportunity to consider this manuscript and look forward to your revision.

Referee #1:

The manuscript by Ohnstad et al. reports a role for the cargo receptor TAX1BP1 in the autophagic delivery of NBR1 into lysosomes in ATG7 deficient cells. This role of Tax1BP1 is independent of its ubiquitin binding domains but depends on its ability to bind NBR1 and RB1CC1. These interactions help to link the early autophagy machinery to NBR1 thereby to locally induce autophagosome formation.

The manuscript reports a number of insights that are interesting for the autophagy community. However, there are several issues that make it less exciting for a wider audience.

1. It is unclear what the physiological relevance of ATG7-independent autophagy is. While it is evident from various previous publications that in mammalian cells autophagy including macroautophagy can occur in the absence of ATG7 and LC3 lipidation (see for example PMIDs: 27885029, 27864321), it is unclear if ATG7 independent autophagy occurs in wild type cells. In other words, do the discoveries in this manuscript inform about the progression of autophagy in normal cells expressing ATG7?

2. The authors write "Our data...highlights the duality of autophagy receptor function, both bridging cargo and membranes (via LC3) and/or recruiting upstream autophagy factors to drive cargo-driven autophagosome formation". However, this concept is now well established. See some recent examples PMIDs: 30853401, 30853402, 30853400, 29290589, 27768871, 26166702. Many of the papers are also cited in the text.

3. The authors claim to define a ubiquitin-independent mode of TAX1BP1 recruitment to cargo. However, they show that TAX1BP1 is in a complex with NBR1 and possibly p62, which are ubiquitin binding proteins. Therefore, this claim is not supported by the presented evidence.

Referee #2:

In this manuscript the authors describe a role for TAX1BP1 in mediating autophagic clearance of NBR1 in the absence of Atg7 (and other Atg8 conjugation machinery components). This activity of TAX1BP1 is dependent on direct binding to NBR1 via a newly identified N-domain and via binding and clustering of FIP200 (RB1CC1) via TAX1BP1's SKICH domain. This mechanism of TAX1BP1 activity, which functions independently of ubiquitin binding, drives robust turnover of NBR1 in the absence of Atg8 family lipidation. Overall, the data are mostly clear and convincing and supported by well controlled experiments. The discovery of lipidation and ubiquitin independent autophagy receptor activity is a very interesting phenomenon that will be of interest to many in the autophagy field and also to the broad readership of EMBO J. However, it will be critical for the authors to provide some functional insight into what the role of lipidation independent turnover of NBR1 by TAX1BP1 may play and/or under what conditions. This is important to address given that in Figure 5B the TAX1BP1 Δ O-domain mutant which retains its ability to bind NBR1 and drive turnover in Atg7 KO is not itself turned over at all in reconstituted TAX1BP1 KO cells. This indicates that Atg7 independent turnover is not occurring at all under normal conditions. Thus, what is the role of Atg7 independent autophagy of NBR1 by TAX1BP1? Could it be a compensatory artefact of knocking out components of the lipidation machinery? I think it is certainly possible that TAX1BP1's activity might switch under conditions in which LC3/GABARAP conjugation is compromised in order to support the activity of NBR1 to turn over its substrates (or just to turnover NBR1). Some possible suggestions for the authors to explore and help establish some physiological context include;

- 1) The authors show that ubiquitin is present in the NBR1/TAX1BP1 foci, and NBR1 has previously been shown to turn over ubiquitinated proteins and peroxisomes, does this activity persist in Atg7 KO? Is there an increase in ubiquitinated proteins in either TAX1BP1 KO or Atg7/TAX1BP1 KO cells reconstituted with TAX1BP1 NBR1 binding mutants or SKICH mutants?
- 2) Infection of cells with Legionella results in inhibition of LC3/GABARAP dependent autophagy via the bacterial effector RavZ. Does lipidation independent turnover of NBR1 via TAX1BP1 take over in this context to help clear ubiquitinated substrates, peroxisomes or even invading bacteria? The authors could try infecting with Legionella or overexpressing RavZ.
- 3) Oxidative stress has previously been shown to inhibit Atg4 activity which would prevent LC3/GABARAP maturation. Thus, it is possible that TAX1BP1's activity might switch under conditions in which LC3/GABARAP conjugation is compromised by oxidative stress.

Specific comments:

1. Figure 1H: Can the authors show evidence of autophagosome formation using electron microscopy, ideally using CLEM to show that NBR1 is localised to autophagosomes in Atg7 KO?
2. Figure 4a: Conducting mass spec analysis of the co-IP could help to determine whether any other cargoes are associated and therefore give some insight on the specific substrates that might be normally turned over via NBR1/TAX1BP1 lipidation independent autophagy.

3. Many of the microscope images are too dim to assess. E.g. Figure 4B and C: tfNBR1 is too dim to see in this figure in WT cells. NDP52 is also very difficult to see. It is the same case in Figure 1G and 6A and B. Can the authors please address this issue?

4. Figure 6B: It would be beneficial to re-introduce TAX1BP1 mutants -SKICH mutant and delta N mutant to show what affect they have on NBR1 and FIP200 colocalisation. This will help strengthen the conclusions drawn from the co-IP data.

5. Can the authors confirm that unlipidated LC3 and/or GABARAP subfamily members are not recruited to NBR1/TAX1BP1/FIP200 foci?

6. Does rescue of Atg7 KO (or KO lines of other conjugation proteins) with a catalytic inactive mutant still have the same rate of NBR1 turnover? This could help discount the possibility that NBR1 turnover is a compensatory mechanism in response to the loss of a conjugation protein rather than its activity in lipidation.

Minor comments:

1. It is recommend naming RB1CC1 as FIP200 given that the autophagy field largely recognises it by this name.

2. Line 40: Atg8 family proteins not LC3 family proteins. LC3 proteins are a subfamily of the Atg8 family.

3. Line 362: the RB1CC1 complex the authors refer to is typically called the ULK1 complex. FIP200 is a subunit of the ULK1 complex.

4. Fig 3E: it would be better to include the TAX1BP1 blots in the figure despite being knockouts, this would help to further validate that TAX1BP1 KOs were used.

Referee #3:

Here the authors present a comprehensive use of high-throughput screens, and rigorous genetic knock out models to exploit the panel of tandem fluorescent autophagy reporters established in their previous work (Shoemaker et al., 2019). In this manuscript they address the role of lipidation-independent, ie LC3 independent basal "selective" autophagy.

They show that ATG7-independent autophagy in particular as a modulator of NBR1 flux requires ATG proteins RB1CC1 and ATG9A, and is modulated by TAX1BP1. They show that recruitment of TAX1BP1 in a heterotypic receptor complex can induce puncta with STSQM1 and ubiquitin, and the transport of this complex to lysosomes requires ATG9A. Finally, they identify a domain in TAX1BP1 that mediates NRB1 flux, and FIP200 recruitment. Overall the data is well presented and controlled. There are a few points to address in particular concerning the nature of the NRB1 aggregates, in particular the receptor specific conclusions maybe influenced by the presence of p62 and pathways related to the regulation of p62, and NRB1 levels which are controlled by Keap1-Nrf2. This concern should be considered in light of the publication by Sanchez-Martin et al., (EMBO Rep Jan 2020) which shows overexpression of NRB1 creates liquid droplets influenced by STSQM1/p62

levels and the transcription factor system Keap1-Nrf2. The authors conclusion that a heterotypic receptor substrate assembles as an ATG7 independent system maybe complicated by the formation of liquid droplets in the cells, and the presence of multiple other autophagy receptors. Finally, can this system really be considered as selective autophagy? Or aggrephagy? Given there are just overexpressed receptors and no cargo? Could you comment if TAX1BP1 role described here is associated with a specific cargo degradation (mitophagy, aggrephagy, etc) or is it unspecific?

Specific points:

1. The introduction section is easy to read and very well focused on autophagy and more precisely in selective autophagy in cells lacking LC3 proteins. However, as the main focus of the paper is the role of TAX1BP1 on NBR1 puncta in "selective" autophagy, it would be helpful to include there a brief description of TAX1BP1 and NBR1, covering also their previous described roles in this context.
2. In general, the methods/techniques used in this manuscript are exactly the same used in the previous publication (Shoemaker et al, 2019), missing some novelty in the approach used for the characterization of this new role for TAX1BP1. Moreover, Figure 1 and the first set of results reflect exactly the same results published in Shoemaker et al, 2019. The data about NBR1 and NDP52 degradation upon RB1CC1 depletion is not consistent with data described in Mejlvang et al, 2018.
3. Figure 1D does not reflect the same data if compared with Shoemaker et al, 2019 Figure 2B, where sgATG7 have a strongest effect in autophagy suppression using TAX1BP1 as a reporter. Here in Figure 1D, the values are higher for tf-LC3, tf-NDP52 and tf-SQSTM1 compared to tf-TAX1BP1. Can the authors comment?
4. In Fig S1 there appear large aggregates of NBR1 in FIP200 KO, which is less but noticeable in ATG9A KO. Are these aggregates able to be resolved or refractile to delivery?
5. Can the authors rule out that in ATG9A and FIP200 KO, the capacity to degrade NRB1 is overwhelmed and the aggregates are not engulfed? Levels of NRB1 are increased in ATG9A KO, and should also be controlled in Fip200 KO.
6. A rescue experiment to see test if aggregates in Figure 1G and S1C can be completely degraded should be tried, for example, ie in a stable KO of Fip200 perform an acute rescue and monitor degradation? LC3 could be a positive control.
7. Possibility that NRB1 aggregates are intractable to degradation once formed in lipidation-deficient cells is supported by increased levels seen in Fig. 3C. To bring this in line with the other data please also perform labelling with FIP200 in addition to ATG9A.
8. As shown in Figure 5, ubiquitin binding domains in TAX1BP1 are dispensable for ATG7-independent autophagy. However, is ubiquitin still needed during this process? From these results, it is clear that ubiquitin binding to TAX1BP1 C-terminal region does not occur, but is ubiquitination of cargo still required? Is ubiquitin accumulated in these NBR1 structures in ATG7 KO/TAX1BP1 KO. If so, could you comment on the possible mechanism?
9. Although, as described here LC3 lipidation is not required for NBR1 degradation, are LC3/GABARAPs still essential in this process (non-lipidated forms)? In Figure 5, mutation of non-canonical LIR motif of TAX1BP1 does not affect the rescue experiments in ATG7 KO/TAX1BP1 KO cells, but is the recruitment of LC3/GABARAPs to these NBR1 puncta still taking place?
10. In Figure 5C, coimmunoprecipitation was performed using myc-TAX1BP1 or myc-NBR1 and different tagBFP-V5-TAX1BP1 truncations. Although the result is quite clear, it would be useful to see if these truncations can oligomerize with endogenous TAX1BP1 and/or interact with endogenous NBR1.
11. In Figure 6C, WB or IF analysis of ATG9 depletion in these cell lines should be provided.
12. Could you comment on the role of TAX1BP1 in immediate autophagic response described in Mejlvang et al, 2018? As it is described as an ATG7-independent process, could TAX1BP1 be

playing a role in this situation?

Minor points:

1. Fig. S4 STSQM1 should be included in the IP.
2. Minor spelling corrections:
 - Line 238: Figure S5C
 - Line 262: TAX1BP11-506 Δ O domain
 - Line 299: this interaction is not required
 - Line 332: independent

Referee #1:

The manuscript by Ohnstad et al. reports a role for the cargo receptor TAX1BP1 in the autophagic delivery of NBR1 into lysosomes in ATG7 deficient cells. This role of Tax1BP1 is independent of its ubiquitin binding domains but depends on its ability to bind NBR1 and RB1CC1. These interactions help to link the early autophagy machinery to NBR1 thereby to locally induce autophagosome formation.

The manuscript reports a number of insights that are interesting for the autophagy community. However, there are several issues that make it less exciting for a wider audience.

1. It is unclear what the physiological relevance of ATG7-independent autophagy is. While it is evident from various previous publications that in mammalian cells autophagy including macroautophagy can occur in the absence of ATG7 and LC3 lipidation (see for example PMIDs: 27885029, 27864321), it is unclear if ATG7 independent autophagy occurs in wild type cells. In other words, do the discoveries in this manuscript inform about the progression of autophagy in normal cells expressing ATG7?

Response: These are key questions.

First, what do our data tell us about autophagy? By stripping away the requirement for LC3 lipidation in autophagy, we were afforded an unfiltered view of the underlying mechanisms of cargo selection and autophagosome initiation. These data reinforce the critical role receptors play in clustering early autophagy factors and challenges the primacy of LC3 (although this is not to say that LC3 doesn't make many significant contributions).

Do ATG7-dependent and ATG7-independent autophagy occur independently in WT cells? It is conceptually possible, but with so many interactions between ATGs (in this case, the interaction between FIP200 and ATG16L1 comes to mind), it seems most likely that these are concomitant mechanisms that positively reinforce autophagosome formation around cargo. By necessity, ATG7-independent mechanisms become the predominant form in cases where lipidation is inhibited, (e.g. by bacterial effectors proteins like RavZ [see Fig 4A], or in other disease states).

As a practical matter, our data provide further evidence that common autophagy controls (e.g. *ATG7^{KO}* cells) are insufficient as stand-alone controls for autophagy. This is particularly relevant for members of the broader scientific community who may only choose one or two autophagy controls to test the effects of autophagy in their system of choice.

Finally, our data should inform the clinical development of lipidation machinery inhibitors (e.g. ATG4 or ATG7 inhibitors), which will need to account for the differential degradation of etiological substrates in the presence and absence of lipidation.

2. The authors write "Our data...highlights the duality of autophagy receptor function, both bridging cargo and membranes (via LC3) and/or recruiting upstream autophagy factors to drive cargo-driven autophagosome formation". However, this concept is now well established. See some

recent examples PMIDs: 30853401, 30853402, 30853400, 29290589, 27768871, 26166702. Many of the papers are also cited in the text.

Response: We apologize for our word choice. Our intention was simply to point out that our work reinforces previous findings. We apologize if it seemed as if we were trying to supersede prior claims. We removed this passage from the abstract so as to not over-emphasize this point and to focus on the more novel aspects of our work. We modified (but did not fully eliminate) the related passage in the discussion in order to draw comparisons between this study and previous studies.

In addition, we apologize for the missing reference to Smith et al. (PMID 29290589). We thank the reviewer for pointing out this glaring oversight.

3. The authors claim to define a ubiquitin-independent mode of TAX1BP1 recruitment to cargo. However, they show that TAX1BP1 is in a complex with NBR1 and possibly p62, which are ubiquitin binding proteins. Therefore, this claim is not supported by the presented evidence.

Response: Thank you for pointing out the ambiguities in our claims. All three reviewers made comments related to this point. While the feedback varied, it was clear our comments relating TAX1BP1 and ubiquitin needed to be clarified.

The reviewer is absolutely correct: ubiquitin is present at NBR1 puncta (Data presented as EV4B). A better choice of words could have been “ubiquitin-independent mode of TAX1BP1 recruitment to NBR1” or a “ubiquitin-binding domain (UBZ)- independent mode of TAX1BP1 recruitment to cargo”.

The point we failed to properly convey is that TAX1BP1 functions in ATG7-independent autophagy independent of its ability to bind ubiquitin, which is quite striking. Even though NBR1 puncta have ubiquitin, TAX1BP1 is recruited through a separate mechanism. To clarify this point, we have altered the text of our manuscript to more precisely indicate a “UBZ-independent” recruitment.

In this manuscript the authors describe a role for TAX1BP1 in mediating autophagic clearance of NBR1 in the absence of Atg7 (and other Atg8 conjugation machinery components). This activity of TAX1BP1 is dependent on direct binding to NBR1 via a newly identified N-domain and via binding and clustering of FIP200 (RB1CC1) via TAX1BP1's SKICH domain. This mechanism of TAX1BP1 activity, which functions independently of ubiquitin binding, drives robust turnover of NBR1 in the absence of Atg8 family lipidation. Overall, the data are mostly clear and convincing and supported by well controlled experiments. The discovery of lipidation and ubiquitin independent autophagy receptor activity is a very interesting phenomenon that will be of interest to many in the autophagy field and also to the broad readership of EMBO J. However, it will be critical for the authors to provide some functional insight into what the role of lipidation independent turnover of NBR1 by TAX1BP1 may play and/or under what conditions. This is important to address given that in Figure 5B the TAX1BP1 Δ O-domain mutant which retains its ability to bind NBR1 and drive turnover in Atg7 KO is not itself turned over at all in reconstituted TAX1BP1 KO cells. This indicates that Atg7 independent turnover is not occurring at all under normal conditions. Thus, what is the role of Atg7 independent autophagy of NBR1 by TAX1BP1? Could it a compensatory artefact of knocking out components of the lipidation machinery? I think it is certainly possible that TAX1BP1's activity might switch under conditions in which LC3/GABARAP conjugation is compromised in order to support the activity of NBR1 to turn over its substrates (or just to turnover NBR1). Some possible suggestions for the authors to explore and help establish some physiological context include;

- 1) The authors show that ubiquitin is present in the NBR1/TAX1BP1 foci, and NBR1 has previously been shown to turn over ubiquitinated proteins and peroxisomes, does this activity persist in Atg7 KO? Is there an increase in ubiquitinated proteins in either TAX1BP1 KO or Atg7/TAX1BP1 KO cells reconstituted with TAX1BP1 NBR1 binding mutants or SKICH mutants?
- 2) Infection of cells with Legionella results in inhibition of LC3/GABARAP dependent autophagy via the bacterial effector RavZ. Does lipidation independent turnover of NBR1 via TAX1BP1 take over in this context to help clear ubiquitinated substrates, peroxisomes or even invading bacteria? The authors could try infecting with Legionella or overexpressing RavZ.
- 3) Oxidative stress has previously been shown to inhibit Atg4 activity which would prevent LC3/GABARAP maturation. Thus, it is possible that TAX1BP1's activity might switch under conditions in which LC3/GABARAP conjugation is compromised by oxidative stress.

Response: We are highly appreciative of the reviewer's thoughtful suggestions on how to approach this question. In the end, we settled on using RavZ to mimic a physiological condition in which LC3 lipidation is inhibited and TAX1BP1's activity must "switch." This data is now included in Fig 4A. Related to points 1 and 3, we had issues recapitulating previously published phenomena, so we could not evaluate the effects of TAX1BP1 in those contexts.

Specific comments:

1. Figure 1H: Can the authors show evidence of autophagosome formation using electron microscopy, ideally using CLEM to show that NBR1 is localised to autophagosomes in Atg7 KO?

Response: Thank you for the suggestion. We have performed several CLEM experiments that we believe substantially strengthen our findings. First, we performed CLEM on *ATG7^{KO}* cells and confirmed the localization of NBR1 in double-membrane structures (Fig 1E, EV1E). Second, we performed CLEM on *ATG7^{KO}/TAX1BP1^{KO}* cells and found that NBR1 puncta are no longer surrounded by membrane structures (Fig 3C). We note that NBR1 structures in *ATG7^{KO}/TAX1BP1^{KO}* cells are reminiscent of p62 aggregates recently reported by Jakobi *et al*, 2020. The association of NBR1 with SQSTM1 *in vivo* is consistent with our immunofluorescence microscopy and co-IP data (e.g. Fig 5A and B).

2. Figure 4a: Conducting mass spec analysis of the co-IP could help to determine whether any other cargoes are associated and therefore give some insight on the specific substrates that might be normally turned over via NBR1/TAX1BP1 lipidation independent autophagy.

Response: We performed this analysis as suggested, with mixed results. The data validate that NBR1, TAX1BP1, and SQSTM1 are primary components of this aggregate (this data is now included as Table EV3). The only other known autophagy receptors we observed in this complex were NIPSNAP1 and NIPSNAP2. However, these data did not validate via immunoblotting. We also identified (and validated) TROVE2 as co-IPing with NBR1 in a highly specific manner (Fig EV3B). However, total cellular TROVE2 was not influenced by autophagy (Fig EV3C). While this does not eliminate the possibility that TROVE2 is an autophagy target, we cannot definitively describe the cargo at this time.

3. Many of the microscope images are too dim to assess. E.g. Figure 4B and C: tfNBR1 is too dim to see in this figure in WT cells. NDP52 is also very difficult to see. It is the same case in Figure 1G and 6A and B. Can the authors please address this issue?

Response: We apologize for the poor image quality. It was particularly egregious in print format. We have made several changes to improve the quality of our images. First, we remade all microscopy figures, linearly adjusting brightness and contrast (if possible) and exporting figures in a different file format that improved image quality. Unfortunately, due to intensity differences between cell lines, this could not fully correct the problem. Particularly in cells with high autophagic flux, NBR1 signal is often extremely low. To further improve readers' ability to evaluate our microscopy data, we transitioned to using black/white images for all single-color panels. This allows for a better visualization of intensity in both electronic and print formats. Finally, we note that, in conditions where puncta were not observed, representative cytoplasmic regions were selected to showcase diffuseness of signal (e.g. Fig 5B, receptors are diffuse in WT cells, but accumulate strikingly upon ATG9A KO).

4. Figure 6B: It would be beneficial to re-introduce TAX1BP1 mutants -SKICH mutant and delta N mutant to show what affect they have on NBR1 and FIP200 colocalisation. This will help strengthen the conclusions drawn from the co-IP data.

Response: This is a great suggestion. First, the disappointing news: Simultaneously performing the rescue experiment and immunofluorescence proved technically challenging. Due to sample loss inherent in our immunofluorescence protocol (there are myriad pelleting steps since these are suspension cells), we could never find more than a few BFP-positive cells by the end and we couldn't make meaningful comparisons. However, (the good news) we were able to perform aspects of this suggestion without the immunofluorescence component. With this approach, we were able to validate a role for the N domain in localizing TAX1BP1 to NBR1 puncta *in vivo*. These data are now provided in Fig EV4D, which nicely complement the conclusions from our co-IP data.

5. Can the authors confirm that unlipidated LC3 and/or GABARAP subfamily members are not recruited to NBR1/TAX1BP1/FIP200 foci?

Response: This concern was also shared by Reviewer 3 (point 9). This point is particularly relevant given work by Runwal et al (PMID 31300716) demonstrating that SQSTM1 aggregates can recruit non-lipidated LC3B. To test whether soluble LC3B is similarly recruited to NBR1 aggregates in our system, we performed immunofluorescence microscopy for LC3A/B in various deletion backgrounds. However, we observe no LC3A/B puncta in any cell line lacking the lipidation machinery. These data are now included as Fig EV2E.

6. Does rescue of Atg7 KO (or KO lines of other conjugation proteins) with a catalytic inactive mutant still have the same rate of NBR1 turnover? This could help discount the possibility that NBR1 turnover is a compensatory mechanism in response to the loss of a conjugation protein rather than its activity in lipidation.

Response: This is an important potential caveat. To directly test this alternative hypothesis, we transfected *ATG7^{KO}/TAX1BP1^{KO}* cells with wild-type (BFP-tagged) ATG7 or a catalytically dead variant (BFP-ATG7^{C572S}). As expected, only WT ATG7 rescued; catalytically dead ATG7 had no effect. These data are now included as Fig EV2C. In addition, we note new data from RavZ-expressing cells (Fig 4A). This further distinguishes between LC3 lipidation and ATG7 activity since RavZ inhibits lipidation while leaving the lipidation machinery intact.

Minor comments:

1. It is recommend naming RB1CC1 as FIP200 given that the autophagy field largely recognises it by this name.

Response: We have changed the nomenclature throughout the manuscript to consistently reference FIP200.

2. Line 40: Atg8 family proteins not LC3 family proteins. LC3 proteins are a subfamily of the Atg8 family.

Response: Thank you for pointing out this mistake in nomenclature. We have now clarified “Atg8-family proteins, including both the LC3 and GABARAP families in mammals.”

3. Line 362: the RB1CC1 complex the authors refer to is typically called the ULK1 complex. FIP200 is a subunit of the ULK1 complex.

Response: Thank you for pointing out this ambiguity. We have removed the reference to a complex and refer only to FIP200 (since this is the protein we were directly visualizing).

4. Fig 3E: it would be better to include the TAX1BP1 blots in the figure despite being knockouts, this would help to further validate that TAX1BP1 KOs were used.

Response: This data is now included. We note that this is now Fig 4D.

Referee #3:

Here the authors present a comprehensive use of high-throughput screens, and rigorous genetic knock out models to exploit the panel of tandem fluorescent autophagy reporters established in their previous work (Shoemaker et al., 2019). In this manuscript they address the role of lipidation-independent, ie LC3 independent basal "selective" autophagy.

They show that ATG7-independent autophagy in particular as a modulator of NBR1 flux requires ATG proteins RB1CC1 and ATG9A, and is modulated by TAX1BP1. They show that recruitment of TAX1BP1 in a heterotypic receptor complex can induce puncta with STSQM1 and ubiquitin, and the transport of this complex to lysosomes requires ATG9A. Finally, they identify a domain in TAX1BP1 that mediates NBR1 flux, and FIP200 recruitment. Overall the data is well presented and controlled. There are a few points to address in particular concerning the nature of the NBR1 aggregates, in particular the receptor specific conclusions maybe influenced by the presence of p62 and pathways related to the regulation of p62, and NBR1 levels which are controlled by Keap1-Nrf2. This concern should be considered in light of the publication by Sanchez-Martin et al., (EMBO Rep Jan 2020) which shows overexpression of NBR1 creates liquid droplets influenced by STSQM1/p62 levels and the transcription factor system Keap1-Nrf2. The authors conclusion that a heterotypic receptor substrate assembles as an ATG7 independent system maybe complicated by the formation of liquid droplets in the cells, and the presence of multiple other autophagy receptors. Finally, can this system really be considered as selective autophagy? Or aggrephagy? Given there are just overexpressed receptors and no cargo? Could you comment if TAX1BP1 role described here is associated with a specific cargo degradation (mitophagy, aggrephagy, etc) or is it unspecific?

Response:

We thank the reviewer for their comments.

There are a number of really good points to address. First, as the reviewer points out, the interplay between NBR and p62 was deserving of a better look in our system. Included in the current version of the manuscript, we can now confirm that neither deletion nor overexpression of SQSTM1 affects NBR1 flux in our system (data now included in Fig EV3F, G and H). Moreover, we found that NBR1 overexpression does not affect SQSTM1 flux in K562 cells (Fig EV3I). This contrasts with Sanchez-Martin *et al*, in which NBR1 overexpression clearly inhibits autophagy. While the reasons for this discrepancy are unclear, NBR1 overexpression – in our system – does not have an overtly suppressive effect on autophagy. Therefore, we believe that the NBR1/SQSTM1 dynamics reported in Sanchez-Martin *et al* do not significantly change the interpretation of our manuscript.

In addition to the data above, we have done multiple new experiments to further probe the nature of the NBR1 aggregate (see specific points #4-7).

Finally, the question was posed as to which form of autophagy is occurring under these conditions. We confirmed the presence of ubiquitin at these puncta (see specific point #8). This supports the idea that ubiquitinated cargo is likely being degraded, not just “empty” receptors with no cargo. However, to date, the identity of this cargo remains unclear. Neither mass spectroscopy nor CLEM

revealed a consensus target for us to pursue. We are undertaking various efforts to identify the cargo, but we cannot definitively describe the cargo at this time.

Specific points:

1. The introduction section is easy to read and very well focused on autophagy and more precisely in selective autophagy in cells lacking LC3 proteins. However, as the main focus of the paper is the role of TAX1BP1 on NBR1 puncta in "selective" autophagy, it would be helpful to include there a brief description of TAX1BP1 and NBR1, covering also their previous described roles in this context.

Response: We have included a new introductory paragraph (reproduced below). In this paragraph, we more concretely introduce autophagy receptors, which facilitates our discussion of receptor activity later in the manuscript. We thank the reviewer for the suggestion.

“Here, we further explored routes to the lysosome by focusing on the SQSTM1-like family of autophagy receptors, including NDP52, SQSTM1, TAX1BP1 and NBR1 (Birgisdottir *et al*, 2013). SQSTM1-like receptors (SLRs) are soluble, cytosolic proteins that, individually or in combination, bind autophagic cargoes and mark them for degradation. SLRs share at least three defining features: a ubiquitin-binding domain, an LC3-binding motif, and an oligomerization domain (for reviews, see Kirkin & Rogov, 2019; Johansen & Lamark, 2020). Accordingly, SLRs perform related, if not redundant, functions during many forms of selective autophagy, including mitophagy (Lazarou *et al*, 2015), xenophagy (Tumbarello *et al*, 2015), and aggrephagy (Sarraf *et al*, 2019; Kirkin *et al*, 2009). At the same time, receptor diversity enables individual SLRs to perform non-overlapping functions, such as targeting unique autophagy substrates or interfacing with additional cellular pathways. For instance, NBR1 functions coordinately with SQSTM1 in aggrephagy (Kirkin *et al*, 2009) and pexophagy (Deosaran *et al*, 2013), yet uniquely targets alternative substrates (e.g. major histocompatibility complex class I [MHC-I]) (Yamamoto *et al*, 2020). Additionally, NBR1 is one of several receptors that, in response to acute starvation, also becomes a target of endosomal microautophagy (Mejlvang *et al*, 2018).”

2. In general, the methods/techniques used in this manuscript are exactly the same used in the previous publication (Shoemaker *et al*, 2019), missing some novelty in the approach used for the characterization of this new role for TAX1BP1. Moreover, Figure 1 and the first set of results reflect exactly the same results published in Shoemaker *et al*, 2019. The data about NBR1 and NDP52 degradation upon RB1CC1 depletion is not consistent with data described in Mejlvang *et al*, 2018.

Response: This is a point well taken. This had been an attempt to establish previous findings in our own lab. However, they were redundant. We have removed old Fig 1B, 1C, and S1A and moved old Fig 1E to the supplemental data (new Fig EV1A). In its place, we have added new CLEM data (Fig 1E and Fig EV1E) that better illustrate our findings.

There are several possible explanations for the discrepancy between our data and those reported in Mejlvang *et al*, 2018. First, the process reported by Mejlvang *et al* occurs in response to starvation. We purposefully avoided working under starvation conditions so as to not potentially conflate these processes. Second, the FIP200 data from Mejlvang *et al* was generated using siRNA, while we use CRISPR to generate stable knock outs. Mejlvang *et al* assess autophagosome formation (via LC3 puncta) in their siFIP200-treated cells, and they find that autophagosome formation is markedly reduced, but not necessarily absent (Fig S3E in Mejlvang *et al*). In contrast, in our FIP200 KO cells, all LC3 puncta are lost (see data, on right). Interestingly, Mejlvang *et al* test ATG7 function using stable KO cell lines rather than siRNA (Fig 3E, Fig 3F, Fig S3F in Mejlvang *et al*). Perhaps not coincidentally, they see results similar to ours – namely that NBR1 and TAX1BP1 are still degraded, but NDP52 and SQSTM1 are not. These data are fully consistent with our observation that Atg8-lipidation differentially affects receptor flux.

3. Figure 1D does not reflect the same data if compared with Shoemaker *et al*, 2019 Figure 2B, where sgATG7 have a strongest effect in autophagy suppression using TAX1BP1 as a reporter. Here in Figure 1D, the values are higher for tf-LC3, tf-NDP52 and tf-SQSTM1 compared to tf-TAX1BP1. Can the authors comment?

Response: This is an astute observation. We note that the effect sizes in Shoemaker *et al* 2019, Fig 2B, are based on sequencing data, which is prone to noise and may overestimate or underestimate actual effects. However, when we validated the data (Shoemaker 2019, Fig 3B, reproduced on right) the true effect size showed the same trend that we report in this manuscript. Therefore, we believe the data are consistent.

4. In Fig S1 there appear large aggregates of NBR1 in FIP200 KO, which is less but noticeable in ATG9A KO. Are these aggregates able to be resolved or refractile to delivery?

Response: To test whether aggregated NBR1 can be resolved, we electroporated FIP200^{KO} and/or ATG9A^{KO} cells with BFP-FIP200 or BFP-ATG9A and monitored tf-NBR1 flux as acutely as possible. BFP tagging allowed us to clearly distinguish transfected and non-transfected cells. At 8 hours, we found low levels of BFP expression and partial complementation (data on next page). The doubling time of these cells is ~18 hours, suggesting these effects are not simply due to dilution through cell division. We cannot rule out that a subset of aggregates (e.g. really large ones) is refractory to delivery. However, combined with other data (see points 5 and 6, below), we believe that aggregation does not fully render NBR1 refractory to delivery. These data were not included in the new manuscript, because they are very similar to new data from Reviewer comment #6 (which was included in the manuscript).

5. Can the authors rule out that in *ATG9A* and *FIP200* KO, the capacity to degrade NBR1 is overwhelmed and the aggregates are not engulfed? Levels of NBR1 are increased in *ATG9A* KO, and should also be controlled in *Fip200* KO.

Response: To evaluate the effect of NBR1 levels on turnover, we took two approaches. First, we monitored the correlation between tf-NBR1 expression and flux (Fig EV1F). Even at the lowest levels of overexpression, tf-NBR1 is completely refractory to lysosomal delivery in *ATG9A^{KO}* cells. However, this is still an overexpression system. Therefore, we used shRNA to knock down NBR1 expression in *ATG9^{KO}* and *FIP200^{KO}* cells (data above). By this approach, we are able to generate *ATG9^{KO}* and *FIP200^{KO}* cells that express NBR1 at or below the levels observed in WT and *ATG7^{KO}* cells. Even under these conditions, tf-NBR1 flux is fully inhibited. We believe these data indicate that autophagy is truly inhibited, not simply overwhelmed.

6. A rescue experiment to see test if aggregates in Figure 1G and S1C can be completely degraded should be tried, for example, ie in a stable KO of *Fip200* perform an acute rescue and monitor degradation? LC3 could be a positive control.

Response: This was a clever suggestion. To test this hypothesis, we electroporated *FIP200^{KO}* cells (expressing either tf-NBR1 or tf-LC3) with a BFP-FIP200 rescue construct (or a BFP-only control). 8 hours post-transfection, the rescue of tf-NBR1 (an aggregate) and tf-LC3 (a soluble control) was comparable (albeit not complete in either case). These data are now included in Fig EV1D. By 24 hours, flux of both reporters was fully rescued (data not shown). However, because our cells have a doubling time of ~18 hours, we prefer the 8 h timepoint, as it minimizes the possibility that NBR1 aggregates are being diluted by cell division. As noted in our response to comment #4, these data cannot rule out that a subset of aggregates is refractory to delivery. However, we believe these data suggest that aggregation does not fully render NBR1 refractory to delivery.

7. Possibility that NBR1 aggregates are intractable to degradation once formed in lipidation-deficient cells is supported by increased levels seen in Fig. 3C (note from the authors: this is now Fig 4B). To bring this in line with the other data please also perform labelling with FIP200 in addition to ATG9A.

Response: FIP200 and ATG9A are both included in our blot, as requested. The data is presented in Fig EV2F (previously Fig S3E). We are concerned we might not have interpreted this comment correctly. We apologize if we did not perform what the reviewer intended.

8. As shown in Figure 5, ubiquitin binding domains in TAX1BP1 are dispensable for ATG7-independent autophagy. However, is ubiquitin still needed during this process? From these results, it is clear that ubiquitin binding to TAX1BP1 C-terminal region does not occur, but is ubiquitination of cargo still required? Is ubiquitin accumulated in these NBR1 structures in ATG7 KO/TAX1BP1 KO. If so, could you comment on the possible mechanism?

Response: Related questions were posed by all three reviewers, centering on the role of ubiquitin in this process. First, we performed the experiment suggested by this reviewer, looking at colocalization of NBR1 and Ubiquitin (FK2 antibody) in *ATG7^{KO}/TAX1BP1^{KO}* cells. Indeed, these foci are Ub-positive (Data presented as EV4B). A likely model is that that ubiquitinated cargo is recognized by NBR1, which is then recognized by TAX1BP1. This is in line with the observation that the ubiquitin binding domain of NBR1 is important for NBR1 flux (Data presented as EV4C). To further reduce confusion, we have modified the text of our manuscript to more precisely indicate a 'UBZ-independent' mechanism.

9. Although, as described here LC3 lipidation is not required for NBR1 degradation, are LC3/GABARAPs still essential in this process (non-lipidated forms)? In Figure 5, mutation of non-canonical LIR motif of TAX1BP1 does not affect the rescue experiments in ATG7 KO/TAX1BP1 KO cells, but is the recruitment of LC3/GABARAPs to these NBR1 puncta still taking place?

Response: This concern was also raised by Reviewer 2 (point 5). This point is particularly relevant given work by Runwal et al (PMID 31300716) demonstrating that SQSTM1 aggregates can recruit non-lipidated LC3B. To test whether soluble LC3B is similarly recruited to NBR1 aggregates, we performed immunofluorescence microscopy for LC3A/B in various deletion backgrounds. However, we observe no co-staining of LC3A/B and NBR1 in any cell line lacking the lipidation machinery. These data are now included as Fig EV2E.

10. In Figure 5C, coimmunoprecipitation was performed using myc-TAX1BP1 or myc-NBR1 and different tagBFP-V5-TAX1BP1 truncations. Although the result is quite clear, it would be useful to see if these truncations can oligomerize with endogenous TAX1BP1 and/or interact with endogenous NBR1.

Response: This would be a nice addition. Unfortunately, when we tried this approach, we could not detect endogenous NBR1 in the input fraction, suggesting it was below the limit of detection in our HEK293T lysate. In contrast, we could observe endogenous TAX1BP1 in the lysate. However, BFP-TAX1BP1 forms a doublet (see Fig 6C for an example). The lower band migrates in the same region as endogenous TAX1BP1 and is of much higher intensity. This precluded our ability to interpret these experiments. We apologize for not being able to provide these data.

As an alternative approach, we were able to validate the effect of the N domain with fluorescence microscopy (Fig EV4D). Using *ATG9A^{KO}/TAX1BP1^{KO}* cells, we transduced BFP-tagged TAX1BP1 variants and assessed colocalization with tf-NBR1 puncta. As expected, BFP-TAX1BP1¹⁻⁵⁰⁶ coalesced into puncta that colocalized with NBR1, while BFP-TAX1BP1¹⁻⁴²⁰ (which lacks the N domain) remained fully diffuse. These data nicely complement the conclusions from our co-IP data, namely that the second half of CC2 defines a UBZ-independent mode of TAX1BP1 recruitment to NBR1. These data are now provided in Fig EV4D.

11. In Figure 6C, WB or IF analysis of ATG9 depletion in these cell lines should be provided.

Response: These data are now included in Fig EV5B. We apologize for omitting this important control in the original manuscript.

12. Could you comment on the role of TAX1BP1 in immediate autophagic response described in Mejlvang et al, 2018? As it is described as an ATG7-independent process, could TAX1BP1 be playing a role in this situation?

Response: We agree that there is an interesting thread running between these two studies. Particularly, we note the similar pattern of receptor flux in Mejlvang 2018 and this study (namely, the LC3-independent turnover of TAX1BP1 and NBR1, but not NDP52). We anticipate this likely reflects the interaction of TAX1BP1 and NBR1, reported there. At the same time, there are notable differences: starvation vs basal, FIP200-dependent vs independent. Going forward, it will be interesting to use our TAX1BP1 variants to test whether TAX1BP1 is also a key component in their experimental system.

Minor points:

1. Fig. S4 STSQM1 should be included in the IP.

Response: Thanks. This was an important addition to reinforce that this is truly a complex. The IP was repeated and an SQSMT1 blot is now included (new Fig EV3A)

2. Minor spelling corrections:

Response: Thank you for catching these. They have been corrected.

- Line 238: Figure S5C

- Line 262: TAX1BP11-506ΔO domain
- Line 299: this interaction is not required
- Line 332: independent

Thank you for submitting your revised manuscript. The study has been seen by the original referees, whose comments are shown below.

As you can see, the reviewers find that their criticisms have been sufficiently addressed and recommend the manuscript for publication, pending the few minor issues outlined below. In particular, referee #1 requests you to rephrase a statement and to clarify two inconsistencies in the text, whereas referee #3 asks you show the cellular amount of tf-NBR1 compared to endogenous NBR1.

In addition to solving these remaining points, there are a few editorial issues concerning the text and the figures that I need you to address before we can officially accept your manuscript.

Referee #1:

The authors have done an excellent job and the manuscript has been improved substantially. I have only a few comments left the authors should address.

1. In line 304 and following, the authors use tf-TAX1BP1 truncations in TAX1BP1 KO cells and follow the lysosomal delivery of these truncations. They conclude that by this approach they differentiate TAX1BP1 functions in canonical and ATG7-independent autophagy. This wording is somewhat incorrect, because for canonical autophagy, they don't actually assess the function of TAX1BP1 but only its degradation.
2. According to the model in Fig 7F, RB1CC1 is present at the isolation membrane before the cargo is recruited, while in the text the authors clearly state that TAX1BP1 is responsible for RB1CC1 recruitment and clustering at the cargo in LC3-independent autophagy.
3. Line 146 and on: please define better the parameter used for analysis. Are "robustness" and "autophagic efficiency/fractional turnover" the same thing? If so, use only one of the two definition.

Referee #2:

The authors have done an excellent job addressing the comments/suggestions which has strengthened their conclusions and resulted in a very high quality manuscript.

Referee #3:

The authors have nicely addressed all of my comments as well as the cell models they are using allows. The addition of the CLEM data strengthens the conclusion that the structures they observe are indeed either double membrane vesicles or aggregates. The additional data provided in response to the previous points has also clarified some aspects of the study. There remains one point which was not obvious to this reviewer previously, or at least was less important than other concerns. Many of the most important points in the manuscript rely on the behaviour and levels of tf-NBR1. Thus, it is critical to incorporate the knowledge of how much tf-NBR1 is in the cells compared to endogenous protein and to note that this may have an influence on aggregate formation and sequestration dependent or not on autophagy receptors. I understand in the original study (PLOS Biology) the tf-cell lines were designed to express similar amounts for the CRISPR screen. However, here they are used for a different question. It would be important highlight that the level of tf-NBR1 in comparison to the endogenous is significantly higher. It is also clear from Figure 5A that the tf-NBR1 levels increase in particular with the double knockdown of ATG7 and TAX1BP1. If the tf-NBR1 expression levels are increasing with autophagy knockouts can the authors be sure their conclusions about the role of LC3 and FIP200 are not influenced by protein levels? The authors should consider this point in the results text and highlight this possibility in the discussion.

Referee #1:

The authors have done an excellent job and the manuscript has been improved substantially. I have only a few comments left the authors should address.

Response: Thank you for your efforts in helping us to improve this manuscript. Individual responses are indicated below.

1. In line 304 and following, the authors use tf-TAX1BP1 truncations in TAX1BP1 KO cells and follow the lysosomal delivery of these truncations. They conclude that by this approach they differentiate TAX1BP1 functions in canonical and ATG7-independent autophagy. This wording is somewhat incorrect, because for canonical autophagy, they don't actually assess the function of TAX1BP1 but only its degradation.

Response: We have modified this section (lines 311-316) to correct this point. Most notably, "... differentiate TAX1BP1's function in ATG7-independent autophagy from its role as a canonical autophagy substrate."

2. According to the model in Fig 7F, RB1CC1 is present at the isolation membrane before the cargo is recruited, while in the text the authors clearly state that TAX1BP1 is responsible for RB1CC1 recruitment and clustering at the cargo in LC3-independent autophagy.

Response: The preliminary model (Fig 1) and the final model (Fig 7) have been updated.

3. Line 146 and on: please define better the parameter used for analysis. Are "robustness" and "autophagic efficiency/fractional turnover" the same thing? If so, use only one of the two definition.

Response: We amended the text (lines 143-148) to clarify.

Referee #2:

The authors have done an excellent job addressing the comments/suggestions which has strengthened their conclusions and resulted in a very high quality manuscript.

Response: Thank you for your contributions to improving our manuscript.

Referee #3:

The authors have nicely addressed all of my comments as well as the cell models they are using allows. The addition of the CLEM data strengthens the conclusion that the structures they observe are indeed either double membrane vesicles or aggregates. The additional data provided

in response to the previous points has also clarified some aspects of the study. There remains one point which was not obvious to this reviewer previously, or at least was less important than other concerns. Many of the most important points in the manuscript rely on the behaviour and levels of tf-NBR1. Thus, it is critical to incorporate the knowledge of how much tf-NBR1 is in the cells compared to endogenous protein and to note that this may have an influence on aggregate formation and sequestration dependent or not on autophagy receptors. I understand in the original study (PLOS Biology) the tf-cell lines were designed to express similar amounts for the CRISPR screen. However, here they are used for a different question. It would be important to highlight that the level of tf-NBR1 in comparison to the endogenous is significantly higher. It is also clear from Figure 5A that the tf-NBR1 levels increase in particular with the double knockdown of ATG7 and TAX1BP1. If the tf-NBR1 expression levels are increasing with autophagy knockouts can the authors be sure their conclusions about the role of LC3 and FIP200 are not influenced by protein levels? The authors should consider this point in the results text and highlight this possibility in the discussion.

Response: Thank you for your feedback. We have added discussion of these points in the text to outline these potential caveats.

Lines 231, inserted a reference to possible: “tf-NBR1 over-expression or tagging artifacts”.

Line 292, “Since ectopic tf-NBR1 expression increases total NBR1 expression levels (Fig 5A), this may influence receptor dynamics.”

2nd Revision - Editorial Decision

17th Sep 2020

I am pleased to inform you that your manuscript has been accepted for publication in The EMBO Journal.

Congratulations!

Corresponding Author Name: Christopher J Shoemaker

Journal Submitted to: EMBO J

Manuscript Number: EMBOJ-2020-104948